# MicroRNA-21 promotes pancreatic β cell function through modulating glucose uptake

Ruiling Liu[1,2,5], Cuilian Liu[3,5], Xiaozhen He[2], Peng Sun[4], Bin Zhang[2], Haoran Yang [3], Weiyun Shi [2✉] & Qingguo Ruan [2,3✉]

Pancreatic β cell dysfunction contributes to the pathogenesis of type 2 diabetes. MiR-21 has been shown to be induced in the islets of glucose intolerant patients and type 2 diabetic mice. However, the role of miR-21 in the regulation of pancreatic β cell function remains largely elusive. In the current study, we identify the pathway by which miR-21 regulates glucose-stimulated insulin secretion utilizing mice lacking miR-21 in their β cells (*miR-21βKO*). We find that *miR-21βKO* mice develop glucose intolerance due to impaired glucose-stimulated insulin secretion. Mechanistic studies reveal that miR-21 enhances glucose uptake and subsequently promotes insulin secretion by up-regulating Glut2 expression in a miR-21-Pdcd4-AP-1 dependent pathway. Over-expression of Glut2 in knockout islets results in rescue of the impaired glucose-stimulated insulin secretion. Furthermore, we demonstrate that delivery of miR-21 into the pancreas of type 2 diabetic *db/db* male mice is able to promote Glut2 expression and reduce blood glucose level. Taking together, our results reveal that miR-21 in islet β cell promotes insulin secretion and support a role for miR-21 in the regulation of pancreatic β cell function in type 2 diabetes.

[1] School of Basic Medicine, Qingdao University, 266071 Qingdao, People's Republic of China. [2] State Key Laboratory Cultivation Base, Shandong Provincial Key Laboratory of Ophthalmology, Shandong Eye Institute, Shandong First Medical University & Shandong Academy of Medical Sciences, 266071 Qingdao, People's Republic of China. [3] Center for Protein and Cell-Based Drugs, Institute of Biomedicine and Biotechnology, Shenzhen Institutes of Advanced Technology, Chinese Academy of Sciences, 518055 Shenzhen, People's Republic of China. [4] Department of Hepatobiliary and Pancreatic Surgery, The Affiliated Hospital of Qingdao University, 266000 Qingdao, People's Republic of China. [5] These authors contributed equally: Ruiling Liu, Cuilian Liu ✉email: weiyunshi@163.com; ruanqg222@hotmail.com

D iabetes mellitus (DM) is a complex metabolic disorder that can cause blindness, renal failure, and cardiovascular disease. Insulin release from pancreatic β cell is essential for blood glucose homeostasis. In healthy individuals, pancreatic β cell secretes insulin in response to changes in circulating glucose concentration in a process called glucose-stimulated insulin secretion (GSIS). As the most common form of diabetes world-wide, type 2 diabetes (T2D) is a collective state comprising β cell defects in GSIS, impaired glucose tolerance, insulin resistance, and hyperglycemia. As a consequence, this type of diabetes involves defects in both insulin secretion and insulin action[1,2].

MicroRNAs (miRNAs) are a class of ~22-nucleotide-long noncoding RNAs which regulate protein expression at the posttranscriptional level[3]. Increasing number of studies on miRNAs related to human diseases have demonstrated that these molecules play a significant role in controlling cellular pathways. Poy et al. reported the involvement of miRNAs in T2D by showing

that miR-375 plays important roles in insulin secretion[4]. Now many more miRNAs are known to be connected with the cascade of pathways related to T2D. Most miRNAs are involved in insulin secretion because of their abilities to regulate insulin synthesis, ATP/ADP ratio, and insulin granule exocytosis[5]. A few well-studied examples include miR-375, miR-7, miR-124a, miR-9, miR-96, miR-15a/b, miR-34a, miR-195, and miR-376[6].

MiR-21 is one of the most important mammalian microRNAs identified and has been shown to be significantly up-regulated in a wide range of cancers[7]. Because the majority of miR-21 targets are tumor suppressors, the role of miR-21 in apoptosis has been well studied[8–14]. The current study also found that miR-21 is significantly up-regulated in the islets of glucose-intolerant patients and type 2 diabetic mice[15,16]. In addition to the discovery that miR-21 is associated with T2D, it has also been reported that miR-21 may participate in the development of T2D through regulating insulin action and secretion. However, conflicting results were reported when study the role of miR-21 in insulin resistance. While Zhang et al. showed that up-regulation of miR-21 is associated with down-regulated glucose transporter 4 (Glut4) and increased insulin resistance using a rat polycystic ovary syndrome model[17], Ling et al. demonstrated that miR-21 reverses high glucose and high insulin-induced insulin resistance in 3T3-L1 adipocytes through promoting the translocation of Glut4 to plasma membrane[18]. Regarding the role of miR-21 in regulating insulin secretion, Roggli et al. reported that over-expressing miR-21 in MIN6 (a mouse β cell line) cells reduced GSIS[16], which was accompanied by decreased expression of VAMP-2, a SNARE protein that is essential for β cell exocytosis. In this work, we generate mice lacking miR-21 specifically in their β cells (miR-21βKO) and reveal that miR-21 in islet β cell enhances glucose uptake and hence promotes GSIS.

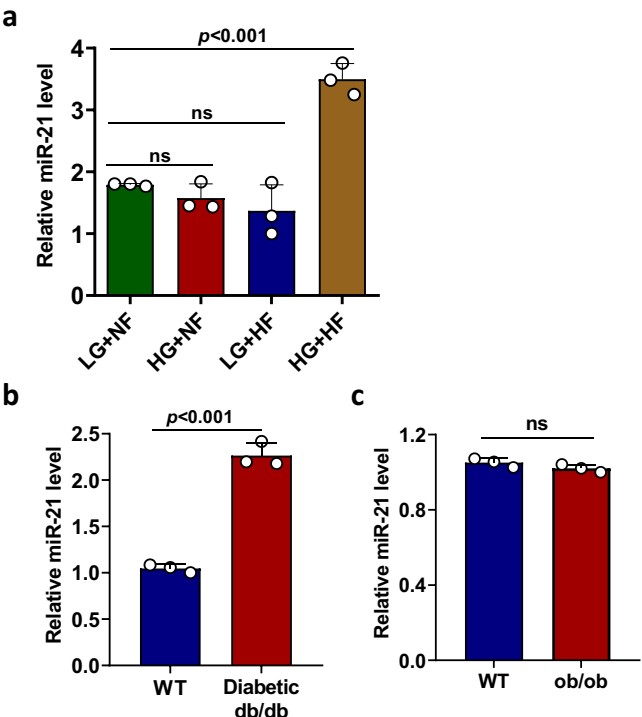

**Fig. 1 Combined high glucose and high fat up-regulates islet miR-21 expression. a** Pancreatic islets were isolated from 8-week-old male C57BL/ 6 mice and treated with or without high glucose and high fat as indicated in the figure. After incubating for 24 h at 37 °C, total RNA was extracted and relative miR-21 expression level was determined by quantitative RT-PCR. Data are presented as means ± SD for n = 3 biologically independent samples. Statistical significance was analyzed using one-way ANOVA with Bonferroni correction. P values are indicated in the figures. ns not significant. LG low glucose (5.5 mM), HG high glucose (25 mM), NF no fat, HF high fat (0.4 mM sodium palmitate). **b** Pancreatic islets were isolated from 6-week-old male diabetic db/db mice and control BKS mice (n = 3). Total RNA was extracted and relative miR-21 expression level was determined by quantitative RT-PCR. Data are presented as means ± SD for n = 3 biologically independent samples. **c** Pancreatic islets were isolated from 20-week-old male ob/ob mice and control C57BL/6 mice (n = 3). Relative miR-21 expression level was determined as in (**b**). Data are presented as means ± SD for n = 3 biologically independent samples. For (**b**) and (**c**), statistical significance was analyzed using two-sided unpaired t test and P values are indicated in the figure. ns not significant. All data shown are representative of three independent experiments. Source data are provided as a Source Data file.

## Results

**Combined high glucose and high fat up-regulates islet miR-21 expression.** Previously it has been shown that islets from glucose-intolerant donors showed significantly higher miR-21 expression than those from healthy donors[15]. Increased islet miR-21 expression has also been reported in islet β cell line and human/mouse islets treated with pro-inflammatory cytokines[16,19]. These indicate that miR-21 in islet β cell may play an important role during the development of type 1 diabetes (T1D). To investigate the role of miR-21 during the development of T2D, we first examined its expression in mouse islets treated in vitro with high glucose, high fat, or a combination of both. Our results showed that mouse islets treated with combined high glucose and high fat but not high glucose or high fat alone significantly up-regulated miR-21 expression (Fig. 1a). These data are consistent with a previous study showing that high glucose alone failed to up-regulate miR-21 expression in INS-1 cells (a rat β cell line)[16]. Next, we examined the expression of miR-21 in glucose and/or toxicity condition in vivo. Our results showed that miR-21 expression in the islets from diabetic db/db mice (Fig. 1b) but not ob/ob mice (Fig. 1c) was significantly increased when compared to control mice. db/db mice display both severe obesity and elevated blood glucose level. However, although ob/ob mice develop severe obesity, they only display mild hyperglycemia. Taken together, we speculate that islet miR-21 expression will be up-regulated under the condition of combined high glucose and high fat but not high fat alone.

**Genetic deletion of miR-21 results in impaired glucose-stimulated insulin secretion.** To investigate the role of miR-21 in regulating the function of pancreatic β cell, we isolated pancreatic islets from wild type (WT) and conventional miR-21-deficient mice (miR-21KO), and cultured them in vitro in the presence of glucose. We found that, while the basal level of

insulin secretion (2.8 mM glucose) was comparable, miR-21-deficient islets showed a significant reduction in insulin secretion when exposed to high concentration (16.7 mM) of glucose (Fig. 2a). To further dissect the role of miR-21 in islet β cell, we generated mice with pancreatic β cell specific deletion of miR-21 using the Cre/Lox system (Fig. S1a). Mutant mice were verified by PCR (Fig. S1b). Homozygous miR-21 floxed mice (f/f) were crossed with Ins2-Cre transgenic animals (Fig. S1c) to selectively ablate miR-21 expression in β cell. Assessment of recombination efficiency by the Cre transgene revealed selective deletion of *miR-21* genes in pancreatic islets (Fig. S1d). Ins2-Cre *miR-21*^f/f mice were born at mendelian frequencies and were seemingly normal. Expression analysis revealed a significant reduction in miR-21 level in Ins2-Cre *miR-21*^f/f (*miR-21βKO*) versus *miR-21*^f/f (WT) pancreas, islets, and islet β cell (Fig. S1e). MiR-21 expression was also reduced in the hypothalamus of *miR-21*KO mice, though not as pronounced as that in the islets. This result is consistent with previous findings that "Ins2-Cre" transgene has been found to be expressed at a low level in the hypothalamus. The expression of miR-21 was comparable in all other tissues examined. Metabolic analysis of mice revealed similar weight (Fig. S2a) and blood glucose (Fig. S2b) in both young (7-week) and old (27-week) *miR-21βKO* mice and control littermates (*miR-21*^f/f). However, glucose tolerance was significantly impaired in 3-, 7-, 10- and 27-week old *miR-21βKO* mice (Fig. 2b) when challenged in an Intraperitoneal Glucose Tolerance Testing (IPGTT). Furthermore, the glucose intolerance in *miR-21βKO* mice was accompanied by reduced insulin concentration in the serum after glucose load (Fig. 2c).

To investigate the cause of impaired glucose tolerance in *miR-21βKO* mice, we analyzed endocrine β cell mass and function. Inspection of islet architecture of young (5-week) and old (27-week) *miR-21βKO* mice revealed intact endocrine cell organization (Fig. S3a). Immunostaining of insulin-positive cells in the islets (Fig. S3b) showed that there was no significant difference in the number of β cell per mm$^2$ islet (Fig. S3c) and islet β cell size (Fig. S3d) between WT and *miR-21βKO* mice. In addition, the percentage of β cell in the islet was also comparable (Fig. S3e). In vitro GSIS assay was performed to further characterize the decreased insulin secretory function of *miR-21βKO* mice. As expected, ex vivo insulin secretion from islets of *miR-21βKO* mice was significantly reduced compared to that of WT mice when islets were exposed to 16.7 mM glucose in vitro (Fig. 2d). However, the mRNA expression of insulin was comparable in WT and *miR-21βKO* islets (Fig. S4), which indicates that miR-21 doesn't affect the biosynthesis of insulin. To exclude the possibility that the observed effects were due to an increase in insulin-producing β cells rather than increased insulin secretion per se, we also examined the fractional insulin release. Our results showed that, while the total insulin content was similar between miR-21-sufficient and deficient islets (Fig. 2e), miR-21-deficient islets exhibited reduced fractional insulin release (Fig. 2f). In addition, to examine the possible off-target effects, we performed a rescue experiment by over-expressing miR-21 in miR-21-deficient islets and then examined glucose-induced insulin secretion. Our results showed that over-expression of miR-21 rescued the impaired glucose-stimulated insulin secretion by islets from *miR-21βKO* mice (Fig. S5).

Obesity is a significant risk factor for the development of glucose intolerance and increases the risk of developing T2D[20,21]. Although *miR-21βKO* mice exhibited impaired glucose tolerance and insulin secretion, the blood glucose level was comparable with that of control mice when mice were fed with a regular diet. To assess the potential role of miR-21 on glucose metabolism under a condition of diabetogenic challenge, WT and *miR-21βKO* mice were fed with high-fat diet for 27 weeks. Our results showed that WT and *miR-21βKO* mice gained the same body weight

(Fig. S6a) and exhibited no significant difference in random blood glucose level (Fig. S6b) and fasting blood glucose level (Fig. S6c). There was also no significant difference in the serum level of triglyceride (TG), total cholesterol (TC), high-density lipoprotein cholesterol (HDLC) and low-density lipoprotein cholesterol (LDLC) between WT and *miR-21βKO* mice (Fig. S6d). However, when a modified IPGTT was performed (mice were fasted for 16 h and then treated with 1 g/kg D-glucose. For regular IPGTT, mice were fasted for 4 h and then treated with 2 g/kg D-glucose), blood glucose intolerance was aggravated in *miR-21βKO* mice (Fig. S7a). An ex vivo insulin secretion assay revealed that islets from *miR-21βKO* mice fed with a high-fat diet also displayed an impaired response to glucose stimulation (Fig. S7b).

Taken together, these results demonstrated that miR-21 in pancreatic islet β cell is a positive regulator of insulin secretion and that its ablation impairs insulin secretion.

**Cellular pathways involved in insulin secretion are impaired in *miR-21βKO* islets.** The principal mechanism of insulin release is GSIS. Glucose is transported into pancreatic β cell by glucose transporter (Glut). ATP produced by glucose metabolism then inhibits ATP-sensitive K$^+$ channels and promotes Ca$^{2+}$ influx. Elevation of Ca$^{2+}$ triggers exocytotic release of insulin granules[22]. To investigate how miR-21 enhances insulin secretion, we performed mRNA sequencing analysis on islets from WT and *miR-21βKO* mice. Among 17,421 genes, we identified 597 up-regulated and 293 down-regulated genes which were statistically significant with P values less than 0.05 (Fig. 3a, b). Gene transcripts that showed significantly different expression profiles were further analyzed for functional Gene Ontology (GO) enrichment. Our results revealed that the types of genes that were differentially expressed clustered into specific functional classes related to insulin secretion. While genes up-regulated in *miR-21βKO* islets clustered into potassium ion transport, negative regulation of glucose import, negative regulation of transmembrane transport, and negative regulation of secretion (Fig. 3b), genes down-regulated clustered into response to glucose, calcium import, calcium-dependent exocytosis, insulin secretion, and intracellular transport (Fig. 3d). Alterations in representative mRNA levels are given in the obtained heatmap (Fig. 3e, f). These data indicated that cellular pathways involved in insulin secretion were impaired in *miR-21βKO* islets.

To identify potential miR-21 targets among the differentially expressed genes, TargetScan was used to search for putative miR-21 binding site in the 3′-UTR of genes that were up-regulated in the islets from *miR-21βKO* mice and also involved in the regulation of insulin secretion. Our results (Supplemental Table 1) showed that 8 out of 33 genes contained potential miR-21 binding in their 3′-UTR. We then used Ago2 immunoprecipitation to further determine whether they are true miR-21 targets. Because it is difficult to obtain a sufficient number of cells from primary islets for Ago2 immunoprecipitation analysis (cells from ~20,000 islets or ~250 mice are needed for one immunoprecipitation analysis), we switched to pancreatic β cell line (MIN6) for this experiment. We first confirmed that the expression of those genes was increased in MIN6 cells transfected with miR-21 antagomir (Fig. S8a). Two genes (Kcnj5 and Scn4a) were not selected for further analysis because their expression in MIN6 cells was barely detectable by RT-PCR. Pdcd4, a well-known miR-21 target, was included in this analysis. Ago2 immunoprecipitation followed by RT-PCR analysis revealed that Ago2-immunoprecipitated RNAs had significantly reduced Pdcd4 and Pten in MIN6 cells transfected with miR-21 antagomir (Fig. S8b), which is consistent with previous findings that both Pdcd4 and Pten are functional targets of miR-21. However, except for androgen-dependent

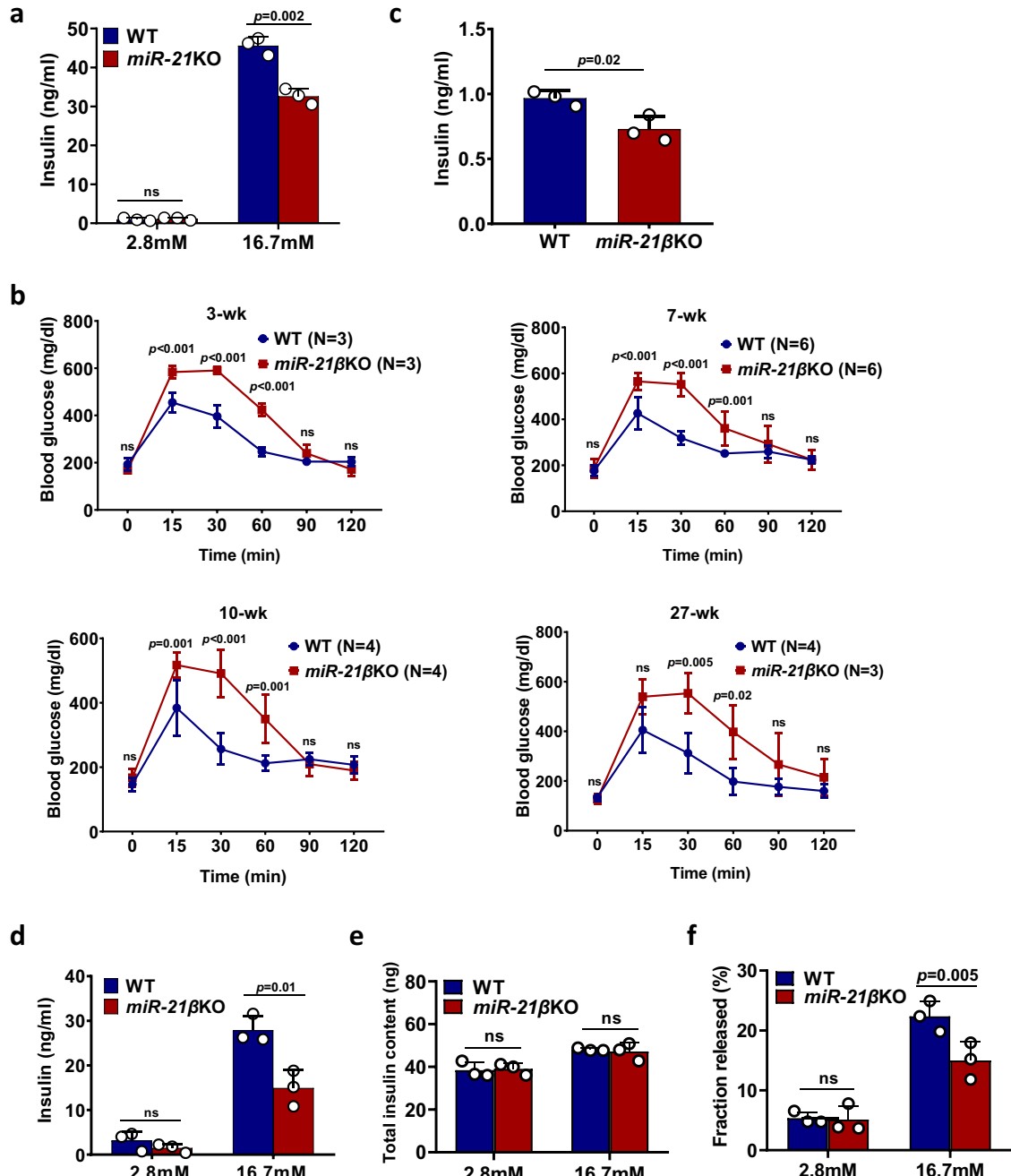

**Fig. 2 Mice with pancreatic β cell specific deletion of miR-21 display impaired blood glucose tolerance and insulin secretion. a** Pancreatic islets from 8-week-old WT and conventional *miR-21*KO mice (*n* = 3) were treated with 2.8 mM or 16.7 mM glucose. Insulin level in the culture supernatant was determined by ELISA. Data are presented as means ± SD for *n* = 3 biologically independent samples. **b** 3-, 7-, 10- and 27-week old wild type (WT) and conditional *miR-21β*KO mice were fasted for 4 h before treated with 2 g/kg body weight D-glucose. Blood glucose level was measured at the indicated time-points. Data are presented as means ± SD for multiple biologically independent mice (as indicated in the figure). Statistical significance was analyzed using two-way ANOVA and *P* values are indicated in the figure. **c** 7-week-old WT and *miR-21β*KO mice (*n* = 3) were treated as in (b) for 30 min. Serum was obtained from tail vein and insulin level was measured. Data are presented as means ± SD for *n* = 3 biologically independent samples. **d–f** Pancreatic islets from 7—8-week old WT and *miR-21β*KO mice (*n* = 3) were treated with 2.8 mM and 16.7 mM glucose. Insulin level in the supernatant was measured (**d**). Alternatively, total islet insulin content (**e**) and fractional insulin release (**f**) from islets treated with 2.8 mM and 16.7 mM glucose were determined by ELISA. Data are presented as means ± SD for *n* = 3 biologically independent samples. All statistical significance analysis except (**b**) was performed using two-sided unpaired *t* test and *P* values are indicated in the figure. All panels except (**b**) are representative of at least two independent experiments. Equal numbers of male and female mice were used in two genotypes. ns: not significant. Source data are provided as a Source Data file.

TFPI-regulating protein (Adtrp), no other genes exhibit a significant difference in Ago2-immunoprecipitated RNAs from MIN6 cells transfected with negative control and miR-21 antagomir (Fig. S8b).

**Islets from *miR-21β*KO mice exhibit impaired glucose uptake.** To further determine which specific pathway was affected by miR-21 deficiency, we first examined insulin secretion evoked by high K[+], an experimental paradigm that bypasses glucose

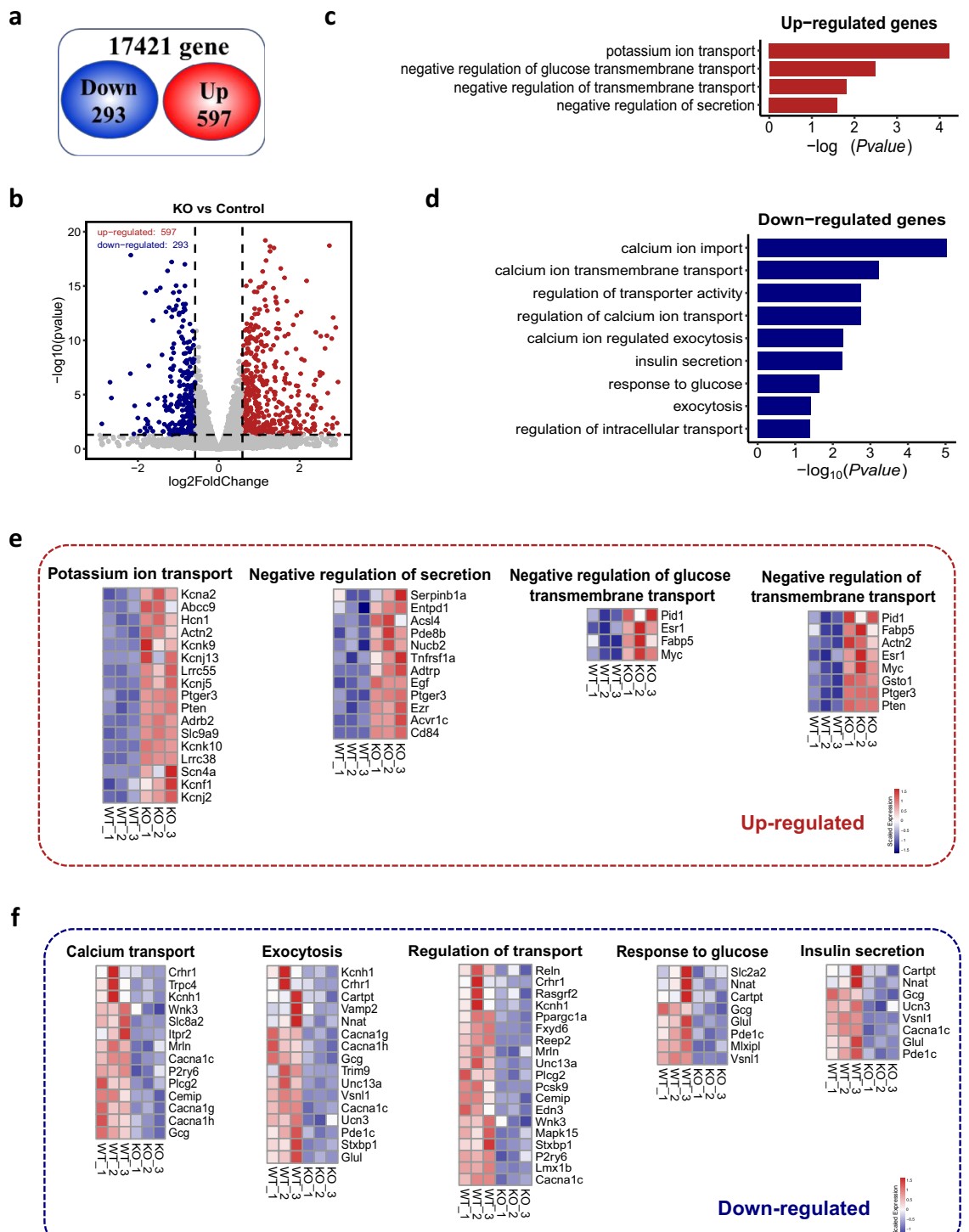

**Fig. 3 Transcriptomic analysis of islets from WT and _miR-21β_KO mice. a** Venn diagram displaying the number of statistically significant altered genes in the islets from _miR-21β_KO mice (_n_ = 6). **b** A volcano plot of differentially expressed genes. The vertical lines represent 1.5-fold increased and decreased expression. The horizontal line shows P = 0.05. Blue and red dots indicate genes with statistically significant differential expression. **c, d** Representative biological processes up-regulated (**c**) and down-regulated (**d**) in the islets from _miR-21β_KO mice were characterized using GO enrichment analysis. **e, f** Representative genes up-regulated (**e**) and down-regulated (**f**) were illustrated using heatmap. Each column represents data from pooled islets of two mice. Equal numbers of male and female mice were used in two genotypes. Source data are provided as a Source Data file.

metabolism. Our results showed that islets from WT and _miR-21β_KO mice produced comparable amount of insulin when treated with 30 mM KCL in vitro (Fig. 4a). These results indicate that miR-21 does not enhance insulin secretion by an effect exerted distally at the level of insulin granule exocytosis. Next,

we examined the production of Glucose 6-phosphate (G6P), a key intermediate in glucose metabolism[23]. Our results showed that islets from _miR-21β_KO mice produced significant less G6P when exposed to high concentration of glucose (Fig. 4b). To further demonstrate that miR-21 plays an important role in

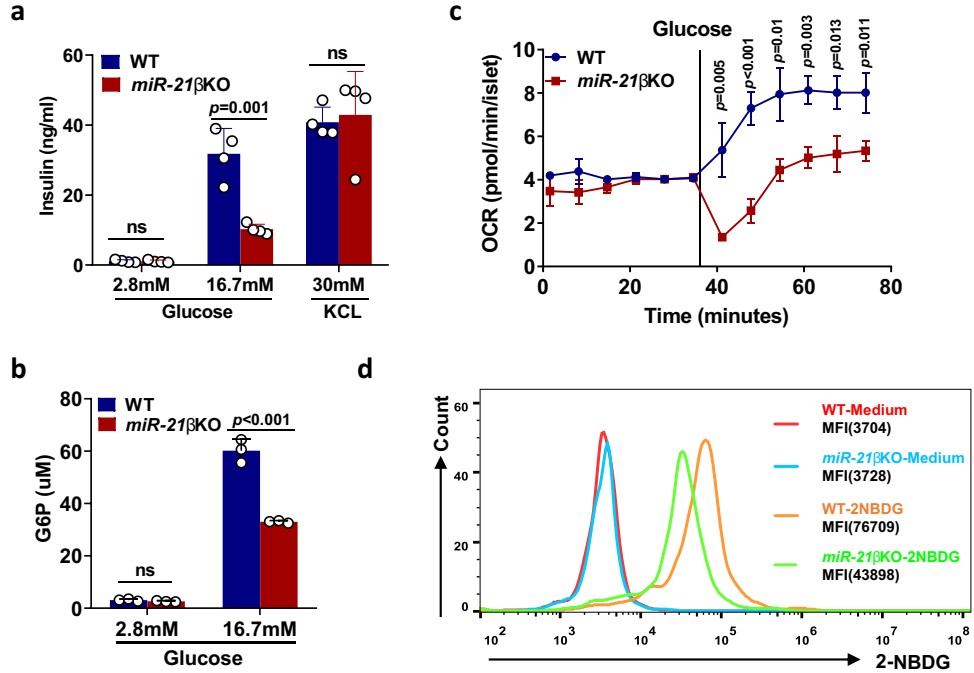

**Fig. 4 Islets from *miR-21β*KO mice exhibit impaired glucose uptake. a** Pancreatic islets isolated from 7–8-week-old WT and *miR-21β*KO mice (n = 4) were treated with indicated concentrations of glucose and KCl. Insulin level in the culture supernatant was determined by ELISA. Data are presented as means ± SD for *n* = 4 biologically independent samples. **b** Pancreatic islets isolated from 7–8-week-old WT and *miR-21β*KO mice (*n* = 3) were treated with indicated concentrations of glucose. Islets were then lysed and G6P level in the lysate was determined using G6P Assay Kit with WST-8. Data are presented as means ± SD for *n* = 3 biologically independent samples. **c** Oxygen consumption rate (OCR) of islets from 7–8-week-old WT and *miR-21β*KO mice (*n* = 3) was assessed as described in the Methods section. Data are presented as means ± SD for *n* = 3 biologically independent samples. Statistical significance was analyzed using two-way ANOVA with Bonferroni correction. *P* values are indicated in the figure. **d** Pancreatic islets from WT or *miR-21β*KO mice (*n* = 3) were cultured at 37 °C for 10 min in the presence or absence of 150 μg/ml NBD-glucose (2-NBDG). Glucose uptake was determined by calculating the mean fluorescence intensity (MFI) of 2-NBDG using flow cytometry. All statistical significance analysis except (c) was performed using two-sided unpaired *t* test and *P* values are indicated in the figure. ns not significant. Data shown are representative of at least two independent experiments. Equal numbers of male and female mice were used in two genotypes. Source data are provided as a Source Data file.

glucose uptake or/and glucose metabolism, we examined oxygen consumption (OCR) when islets were exposed to glucose. The results showed that islets from *miR-21β*KO mice displayed a much reduced response to glucose addition (Fig. 4c). Using glucose uptake assay, we further demonstrated that reduced glucose metabolism by *miR-21β*KO islets was due to defective glucose uptake, which was supported by the decreased intracellular fluorescent intensity of NBD-glucose in *miR-21β*KO islets (Fig. 4d).

**miR-21 improves insulin secretion through up-regulating Glut2 expression**. Glut proteins are a wide group of transmembrane proteins that facilitate the transport of glucose across the plasma membrane[24]. To identify the pathway through which miR-21 promotes glucose uptake, we examined the mRNA expression of five important Gluts in WT and *miR-21β*KO islets stimulated with glucose in vitro. The results showed that Glut2 has the highest expression level in the islets and its expression was significantly decreased in the islets from *miR-21β*KO mice (Fig. 5a). Expression levels of other Gluts were either very low (Glut1, Glut3, and Glut4) or showed no significant difference (Glut5) (Fig. 5a). In addition, we isolated islet β cell from WT and *miR-21β*KO islets and confirmed that Glut2 mRNA expression was decreased in miR-21-deficient β cell (Fig. 5b). We further demonstrated that Glut2 protein expression was significantly decreased in the islets from *miR-21β*KO mice using immunofluorescence staining (Fig. 5c, d) and western blotting (Fig. 5e). In addition, our transcriptomic analysis also confirmed that the

expression of Glut2 (Slc2a2) was reduced in *miR-21β*KO islets (Fig. 3f).

To determine whether decreased Glut2 expression is the cause of defective insulin secretion by *miR-21β*KO islets, we tested whether adenovirally mediated Glut2 transfer could rescue the functional defect observed in the islets from *miR-21β*KO mice. As shown in Fig. 5f, adenovirus infection achieved high and similar efficiency in WT and *miR-21β*KO islets. In addition, islets from WT and *miR-21β*KO mice exhibit no significant difference in apoptosis after adenovirally mediated Glut2-over-expression (Fig. S9). More importantly, ectopic expression of Glut2 in *miR-21β*KO islets resulted in full restoration of the Glut2 mRNA level (Fig. 5g) and rescue of insulin secretion in response to 16.7 mM glucose (Fig. 5h).

Taken together, these results demonstrated that miR-21 promotes insulin secretion through the up-regulation of Glut2 expression.

**miR-21 promotes Glut2 expression through miR-21-Pdcd4-AP-1 pathway**. Previous studies have shown that AP-1 binds to the promoter of Glut2 and positively regulates the Glut2 expression[25]. On the other hand, programmed cell death 4 (Pdcd4), a well studied target of miR-21, is a negative regulator of AP-1 activity[26,27]. To investigate how miR-21 regulates Glut2 expression, we examined whether the expression of Pdcd4 and AP-1 was altered in miR-21βKO islets. Our results showed that, while both mRNA (Fig. 6a) and protein level (Fig. 6b) of Pdcd4 were increased in the islets from miR-21βKO mice,

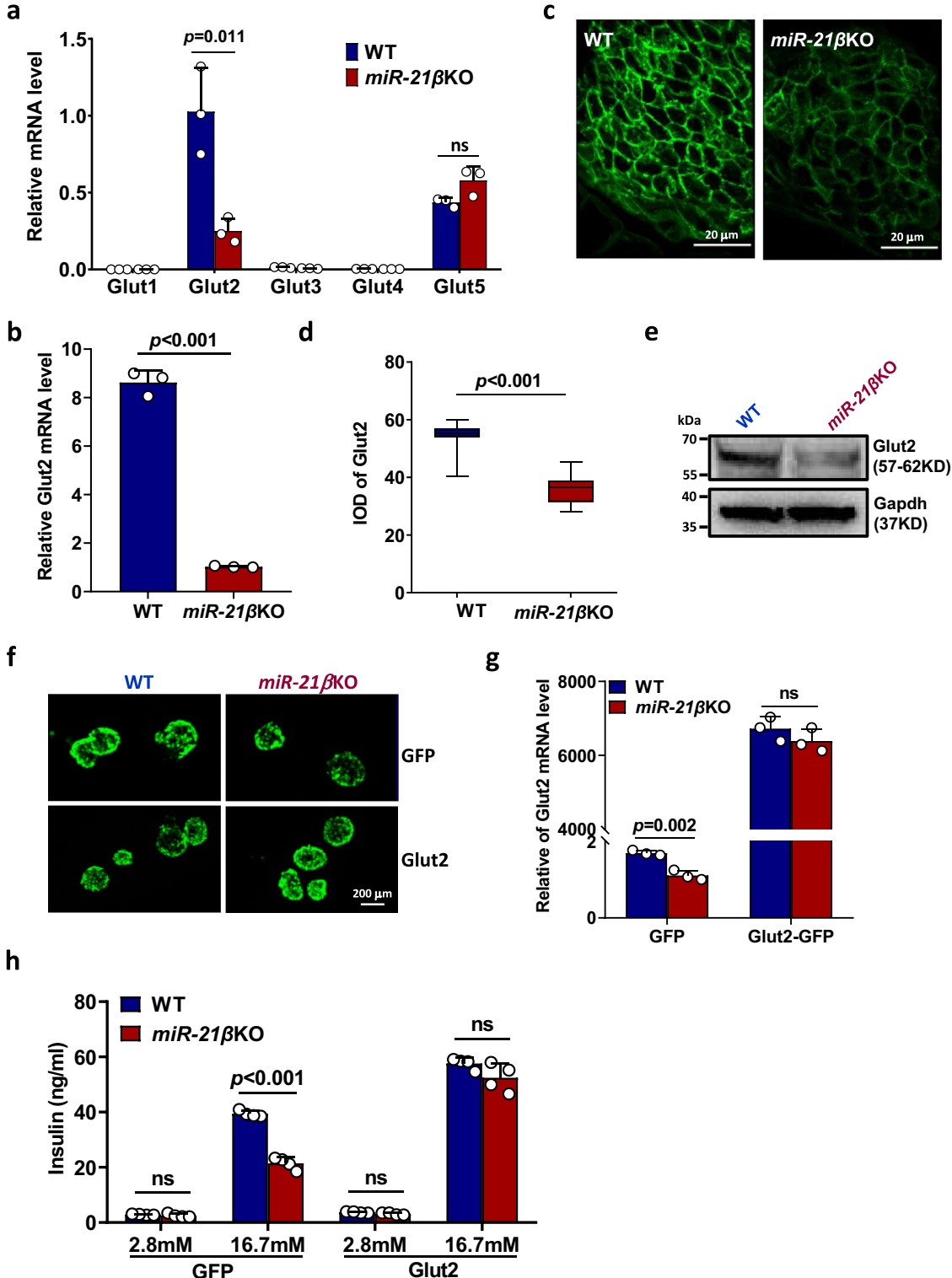

the expression of c-Jun (a subunit of AP-1) was significantly decreased (Fig. 6c).

To test the direct effect of Pdcd4 on Glut2 expression, Pdcd4-specific siRNA was used to knockdown the expression of Pdcd4 in β-TC-6 cells and Glut2 expression was examined by western blotting. We found that, while Pdcd4 expression was efficiently knocked down (Fig. 6d), the protein level of Glut2 was significantly increased (Fig. 6e). To verify that AP-1 regulates the expression of Glut2 in pancreatic β cell, we first constructed a murine Glut2 promoter reporter and co-

transfected with either control or c-Jun-expressing plasmid into β-TC-6 cells. As shown in Fig. 6f, the activity of Glut2 promoter was significantly enhanced when c-Jun was over-expressed. We then examined whether altered AP-1 activity would affect Glut2 expression in pancreatic β cell. Our results showed that, while ectopic over-expression of c-Jun in β-TC-6 cells increased the mRNA (Fig. 6g) and protein level of Glut2 (Fig. 6h), inhibition of AP-1 activity in pancreatic islets with an AP-1 specific inhibitor decreased them (Fig. 6g, i). Collectively, these data indicate that miR-21 in pancreatic β cell may promote

**Fig. 5 miR-21 improves insulin secretion through up-regulating Glut2 expression. a** Pancreatic islets were isolated from 7–8-week-old WT and *miR-21β*KO mice (*n* = 3). Total RNA was extracted and relative mRNA expression levels of indicated Gluts were determined by quantitative RT-PCR. **b** Pancreatic islets were isolated as in (**a**) and islet β cells were further purified by flow cytometry. Glut2 mRNA expression was then determined by quantitative RT-PCR. For both (**a**) and (**b**), data are presented as means ± SD for *n* = 3 biologically independent samples. **c** Representative images of pancreatic tissue sections from 7–8-week-old WT and *miR-21β*KO mice (*n* = 3) incubated with antibody against Glut2. Scale bar: 20 μm. **d** Pancreatic tissues were treated as in (**c**) and mean optical density (MOD) of Glut2 was calculated. Data shown are results combined from 31 islets (WT) or 21 islets (*miR-21β*KO) and presented using box and whisker plot. The line within the box represents the median value. The bottom line of the box represents the 1st quartile. The top line of the box represents the 3rd quartile. The whiskers extend from the ends of the box to the minimum value and maximum value. **e** Pancreatic islets were isolated from 7–8-week-old WT and *miR-21β*KO (*n* = 3) mice, and Glut2 expression was analyzed by western blotting. **f** Representative images of islets from 7–8-week-old WT and *miR-21β*KO mice (*n* = 3) infected with adenovirus expressing negative control (GFP) or murine Glut2 (Glut2). **g** Glut2 mRNA expression in the islets from 7–8-week-old WT and *miR-21β*KO mice (*n* = 3) was determined by quantitative RT-PCR. Data are presented as means ± SD for *n* = 3 biologically independent samples. **h** Adenovirally infected islets from 7–8-week-old WT and *miR-21β*KO mice (*n* = 4) were treated with glucose and insulin level in the culture supernatant was determined by ELISA. Data are presented as means ± SD for *n* = 4 biologically independent samples. All statistical significance was analyzed using two-sided unpaired *t* test and *P* values are indicated in the figure. ns not significant. Results are representative of at least two independent experiments. Equal numbers of male and female mice were used in two genotypes. Source data are provided as a Source Data file.

the expression of Glut2 through miR-21-Pdcd4-AP-1-Glut2 pathway (Fig. 6j).

**Delivery of miR-21 into the pancreas of type 2 diabetic mice reduces blood glucose level.** Since we have established miR-21 as a positive regulator of insulin secretion in islet β cell, next we sought to investigate whether increasing the level of miR-21 could improve insulin secretion and hence reduce the blood glucose level. We first used adenovirus to increase miR-21 level in the islets in vitro. As shown in Fig. 7a, miR-21 level and Glut2 mRNA expression were significantly increased after miR-21 over-expression in mouse islets. More importantly, adenovirally mediated elevation of miR-21 significantly increased glucose-induced insulin secretion (Fig. 7b). Similar results were found when human islets were used (Fig. 7c, d). In addition, we found that adenovirally mediated miR-21 over-expression in mouse islets under the control of mouse insulin-1 promoter also increased Glut2 mRNA expression (Fig. 7e) and glucose-induced insulin secretion (Fig. 7f). To further confirm that over-expression of miR-21 increased insulin secretion per se, we also examined the fractional insulin release. Our results showed that, while total insulin content was similar between control and miR-21-over-expressing islets (Fig. S10a), the fractional insulin release was significantly increased by miR-21-overpressing islets (Fig. S10b).

Next, we sought to determine whether increasing the level of miR-21 in the islets in vivo could reduce blood glucose level. Pancreas in vivo transfection reagent, which is a liposome (cationic lipid) based formulation optimized for in vivo administration with targeted pancreas delivery, was used to specifically increase miR-21 level in the pancreas. Control or miR-21 agomir were administrated through intravenous injection to 6-week-old type 2 diabetic *db/db* mice and body weight/ blood glucose level were measured as illustrated in Fig. 8a. We found that increasing miR-21 level in the pancreas could significantly reduce blood glucose level (Fig. 8b), while body weight (Fig. 8c), percent of body fat (Fig. 8d) and lipid profiles (Fig. 8e) were not affected. We could not directly show that miR-21 level was increased using RT-PCR because miR-21 agomir used has been modified. Instead, we confirmed that miR-21 functions specifically in the pancreas by showing that Glut2 mRNA level was only increased in the pancreas (Fig. 8f). In addition, we further confirmed that Glut2 protein level was also increased in the pancreas (Fig. 8g). In addition, we found insulin concentration in the serum was significantly increased after administration of miR-21 agomir (Fig. 8h). Collectively, these data demonstrated that increasing miR-21 level in the

pancreas is able to promote Glut2 expression and insulin secretion, thereby significantly reducing blood glucose level.

## Discussion

Although miR-21 was up-regulated in the islets of type 2 diabetic *db/db* mice, its role during the development of type 2 diabetes remains unknown. Previously it has been shown that over-expression of miR-21 in MIN6 cells did not significantly affect insulin content, insulin promoter activity or proinsulin mRNA levels[16]. We further confirmed this by showing that the mRNA expression of insulin was comparable in WT and *miR-21β*KO islets. However, compared with previously reported study, conflicting results were generated when studying the function of miR-21 in islet β cells using *miR-21β*KO mice. Roggli et al. reported that over-expression of miR-21 in MIN6 cells reduced glucose-induced insulin release through the down-regulation of VAMP-2, a SNARE protein that is essential for β cell exocytosis. However, utilizing mice with β cell specific deletion of miR-21, we surprisingly found that *miR-21β*KO islets were defective in glucose-induced insulin secretion in vitro, which could be due to decreased glucose uptake by miR-21 deficient islets. Importantly, we found that *miR-21β*KO mice developed glucose intolerance with impaired GSIS in vivo, which is consistent with the results from our in vitro study. Our transcriptomic analysis also revealed that VAMP-2 expression and β cell exocytosis were decreased in *miR-21β*KO islets. However, since we have shown that miR-21 does not enhance insulin secretion by an effect exerted distally at the level of insulin granule exocytosis and over-expression of Glut2 in knockout islets resulted in the rescue of the impaired GSIS, the alteration of the cellular components related to insulin secretion may be secondary to decreased Glut2 expression. Whether the discovery that miR-21 plays opposite role in insulin secretion is related to different cell types used (mouse β cell line *vs* primary mouse islets) or difference in the manipulation of miR-21 expression (over-expression *vs* genetic deletion) remains to be further clarified.

It has been reported that over-expression of miR-21 increased insulin-induced glucose uptake in insulin-resistant adipocytes through promoting the translocation of Glut4 to plasma membrane[18]. We also found that islets from *miR-21β*KO mice exhibit impaired glucose uptake. However, since mouse islets express very little Glut4, we speculate that it is unlikely that miR-21 increases glucose uptake through promoting the translocation of Glut4 to plasma membrane. Instead, our current study suggested that miR-21 may increase glucose uptake by islet β cell through promoting Glut2 expression. On the other hand, since miR-21 promotes rather than reduces Glut2 expression in islet β

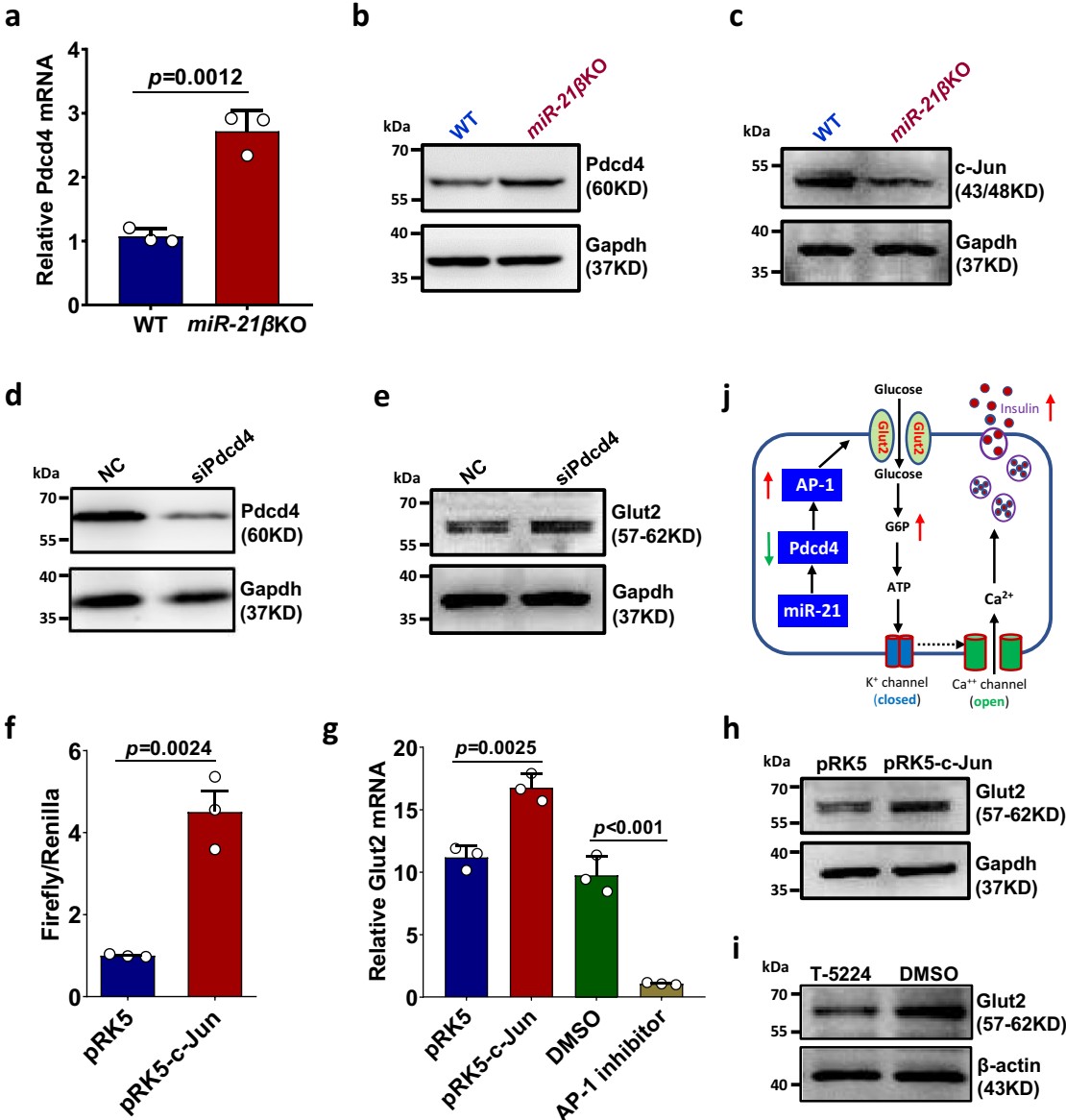

**Fig. 6 miR-21 promotes Glut2 expression through miR-21-Pdcd4-AP-1 pathway. a** Pancreatic islets were isolated from 7–8-week-old WT and *miR-21βKO* mice ($n = 3$). Pdcd4 mRNA expression was determined by quantitative RT-PCR. Data are presented as means ± SD for $n = 3$ biologically independent samples. **b, c** Pancreatic islets were isolated from 7–8-week-old WT or *miR-21βKO* mice ($n = 3$) and protein expression of Pdcd4 (**b**) and c-Jun (**c**) were determined by western blotting. **d, e** β–TC-6 cells were transfected with either universal negative control siRNA duplex (NC) or siRNA duplex targeting mouse Pdcd4 (siPdcd4). Protein levels of Pdcd4 (**d**) and Glut2 (**e**) were examined by western blotting. **f** β–TC-6 cells were transiently transfected with mouse *Glut2* promoter luciferase reporter together with c-Jun expression vector (pRK5-c-Jun) or empty vector (pRK5). The luciferase activities of total cell lysates were measured using the Dual-Luciferase Reporter Assay system. Co-transfection of the Renilla luciferase expression vector pRL-TK was used as an internal control. **g** β–TC-6 cells were transfected with c-Jun expression vector (pRK5-c-Jun) or empty vector (pRK5). Alternatively, β–TC-6 cells were treated with DMSO or AP-1 inhibitor T-5224. Glut2 mRNA was determined by quantitative RT-PCR. For both (**f**) and (**g**), data are presented as means ± SD for $n = 3$ biologically independent samples. **h** β–TC-6 cells were transfected with c-Jun expression vector (pRK5-c-Jun) or empty vector (pRK5). Glut2 protein level was determined by western blotting. **i** Pancreatic islets were treated with DMSO or AP-1 inhibitor T-5524 and Glut2 protein level was determined by western blotting. **j** Schematic illustration of how miR-21 promotes Glut2 expression in islet β cells. All statistical significance was analyzed using two-sided unpaired *t* test and *P* values are indicated in the figure. All results are representative of at least two independent experiments. Equal numbers of male and female mice were used in two genotypes. Source data are provided as a Source Data file.

cell, it is unlikely that Glut2 is a direct target of miR-21. This was further confirmed by the fact that no miR-21 binding site was found within the 3'UTR of Glut2 mRNA. Indeed we demonstrated that miR-21 indirectly promotes Glut2 expression though regulating Pdcd4-AP-1-Glut2 pathway.

Our transcriptomic analysis identified 597 up-regulated genes in the islets from *miR-21βKO* mice. Gene Ontology analysis revealed that 33 of them may be involved in the regulation of

insulin secretion. Using Ago2 immuno-precipition analysis, we further showed that Adtrp could be a new target of miR-21 in the islets. Adtrp was found to be involved in coagulation and thrombosis and negatively regulates extracellular matrix constituent secretion in vascular endothelial cell[28]. Although our current study showed that miR-21 in the islets promotes glucose uptake and insulin secretion through up-regulation of Glut2 expression, whether Adtrp negatively regulates insulin secretion

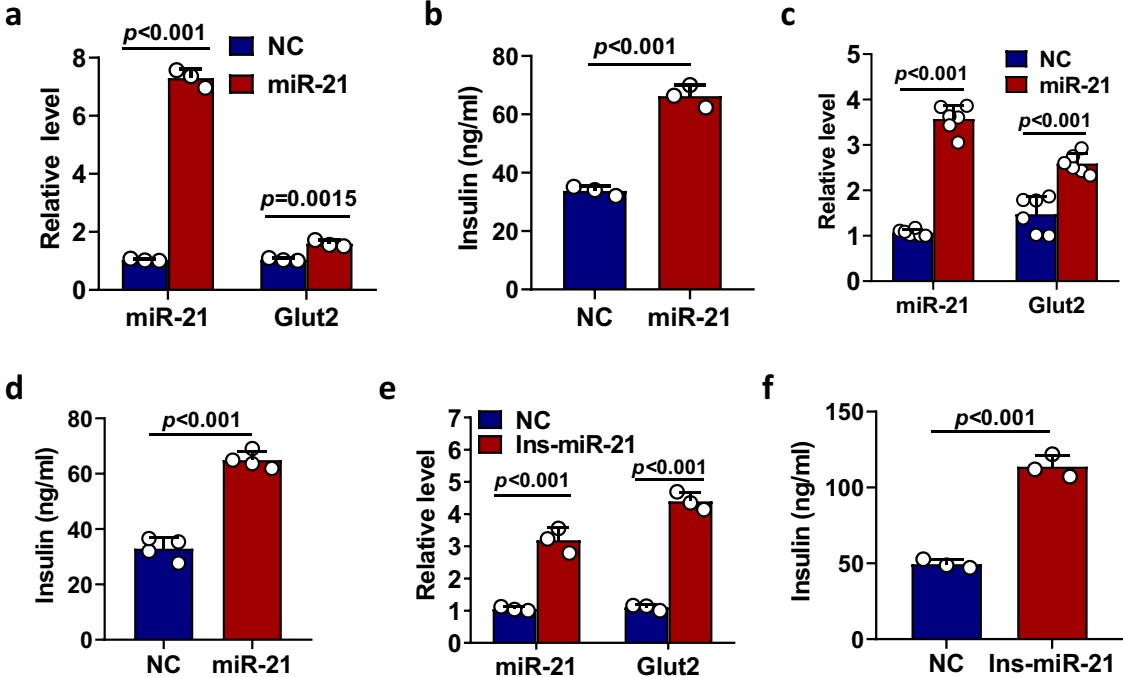

**Fig. 7 Increased expression of miR-21 in pancreatic islets promotes Glut2 expression and glucose-stimulated insulin secretion. a, b** Pancreatic islets were isolated from 7–8-week-old C57BL/6 male mice and infected with negative control virus (NC) or adenovirus over-expressing miR-21 (miR-21). Relative expression levels of miR-21 and Glut2 were determined by quantitative RT-PCR (**a**). Alternatively, virus infected islets were treated with 16.7 mM glucose and insulin level in the culture supernatant was determined by ELISA (**b**). Data are presented as means ± SD for $n = 3$ biologically independent samples. **c, d** Pancreatic islets were isolated from healthy donor and infected with negative control virus (NC) or adenovirus over-expressing miR-21 (miR-21). Expression of human miR-21 and Glut2 (**c**) and the level of glucose-stimulated secretion of insulin (**d**) were determined by quantitative RT-PCR and human insulin ELISA. Data are presented as means ± SD for $n = 3$ biologically independent samples. **e, f** Pancreatic islets were isolated from 7–8-week-old C57BL/6 male mice and infected with negative control virus (NC) or adenovirus over-expressing miR-21 under the control of mouse insulin promoter (Ins-miR-21). Expression of miR-21 and Glut2 (**e**) and the level of glucose-stimulated secretion of insulin (**f**) were determined by ELISA. Data are presented as means ± SD for $n = 3$ biologically independent samples. All statistical significance was analyzed using two-sided unpaired $t$ test and $P$ values are indicated in the figure. ns not significant. Data shown are representative results from at least two independent experiments. Source data are provided as a Source Data file.

in pancreatic islets and whether miR-21 promotes insulin secretion through the inhibition of genes other than Glut2 remains to be further studied.

It has been reported that miRNAs in islet β-cell-derived exosome play an important role in insulin resistance and type 2 diabetes. Indeed miR-21 is enriched in islet β-cell-derived exosome. Although we have shown that miR-21 in islet β cell promotes insulin secretion though Pdcd4-AP-1-Glut2 pathway, whether exosomal miR-21 released by islet β cell regulates insulin resistance and the development of type 2 diabetes warrants further investigation.

Glut2 is mainly expressed in liver, intestine, kidney, central nervous system, and pancreatic islet β cell[29–31]. It is the major glucose transporter in rodent pancreatic β cell. Genetic inactivation of Glut2 has been reported to inhibit glucose uptake and GSIS[24]. Our study revealed that Glut2 expression in murine pancreatic β cell is significantly decreased in the absence of miR-21, which may explain why *miR-21βKO* mice are defective in maintaining proper blood glucose tolerance. However, there are significant differences between rodent and human islets. Numerous studies have shown that GLUT1 and GLUT3 may play an important role in facilitating glucose transportation in human β cell[32]. Therefore, the role of miR-21 in the regulation of GSIS in human islet β cell needs to be determined. Nevertheless, a recent report found that GLUT2 mutations in human cause neonatal diabetes[33], suggesting GLUT2 may also play an important role in human insulin secretion.

Our current study revealed that miR-21 in islet β cell promotes glucose-stimulated insulin release, so the question is why miR-21 level is up-regulated in the islets from diabetic *db/db* mice? We believe that islet miR-21 may act as a negative feedback to high glucose /high fat in diabetic *db/db* mice, and its role is to reduce glucose level through enhancing glucose uptake and insulin release. To test this hypothesis, we increased miR-21 level specifically in the pancreas of diabetic *db/db* male mice using a liposome based transfection reagent that optimized for in vivo administration with targeted pancreas delivery. Our results showed that increasing miR-21 level specifically in the pancreas could significantly reduce blood glucose level. It should be noted that since *db/db* male mice were used in this experimental design, the generalizability to female mice is not known until further investigations. On the other hand, although we have shown that Glut2 expression was elevated only in the pancreas after administration of miR-21 agomir, the specificity of targeted pancreatic β cell delivery cannot be assumed unless more stringent detection methods were used. Even though we have shown that adenovirally mediated miR-21 over-expression in mouse islets under the control of mouse insulin-1 promoter increased glucose-induced insulin secretion, it should be noted that even carrier contains insulin-1 promoter cannot be specifically expressed in islet β-cell. Therefore, the therapeutic translatability of our current study remains to be further investigated.

Although the role of miR-21 in the beta-cell apoptosis has been extensively studied, our current study found that there is

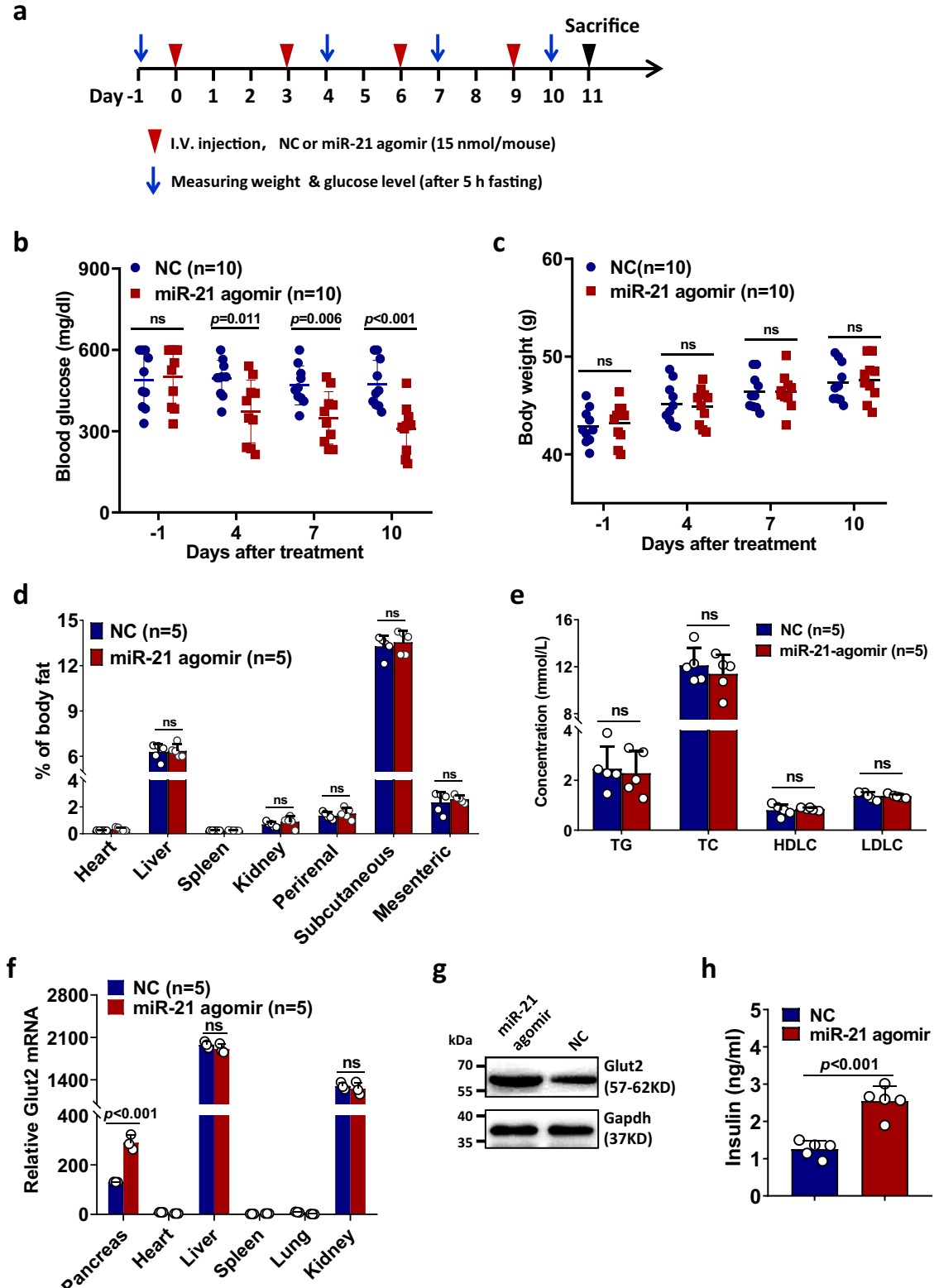

no significant difference in apoptosis between control and miR-21-over-expressed islets. Because the majority of miR-21 targets are tumor suppressors, it is generally believed that miR-21 inhibits apoptosis. However, one group found that increasing miR-21 expression in INS1 cells promotes the apoptosis induced by inflammatory cytokines[12]. Another group reported that inhibiting the expression of miR-21 in MIN6 cells does not affect the apoptosis induced by inflammatory cytokines[16]. We

believe that the above-mentioned inconsistencies may be related to the following factors: 1) apoptosis-inducing conditions; 2) the way miR-21 expression was altered (over-expression or inhibition); 3) types of islet β-cell (cell line or primary islets); 4) the duration of altered miR-21 expression (transient or stable). Nevertheless, since our results showed that adenovirally mediated elevation of miR-21 in mouse pancreatic islets significantly increased glucose-induced fractional insulin release, we believe

**Fig. 8 Delivery of miR-21 into the pancreas of type 2 diabetic *db/db* mice reduces blood glucose level. a** Schematic illustration of the strategy used to treat *db/db* male mice with negative control agomir (NC) or miR-21 agomir. **b**, **c** 6-week-old *db/db* male mice were treated as in (**a**) (10 mice in each group). 5-hour fasting blood glucose level (**b**) and body weight (**c**) were monitored at day −1, 4, 7, and 10. The center line indicates the mean value. **d-h** 6-week-old male *db/db* mice were treated as in (**a**) (5 mice in each group) and sacrificed on day 11 and the percent of body fat of the indicated tissues (**d**), as well as the levels of TG, TC, HDLC and LDLC in the serum (**e**) were measured. Glut2 mRNA level was determined by quantitative RT-PCR (**f**). Data are presented as means ± SD for $n = 5$ biologically independent samples. Alternatively, Pancreatic islets from each group were isolated, mixed and the protein level of Glut2 was detected by western blotting. **g** Insulin concentration in the serum was determined by ELISA (**h**). All statistical significance was analyzed using two-sided unpaired *t* test and *P* values are indicated in the figures. ns not significant. TG triglycerides, TC total cholesterol, HDLC high-density lipoprotein cholesterol, LDLC low-density lipoprotein cholesterol. Results in (**b**) and (**c**) were combined from two independent experiments. All other panels are representative results from two independent experiments. Source data are provided as a Source Data file.

that miR-21 at least partially regulates insulin secretion from pancreatic islet β cell.

In summary, we have demonstrated that miR-21 in pancreatic islet β cell promotes glucose-stimulated insulin secretion and maintains glucose tolerance. Therefore, the current study advanced our understanding of the role of miR-21 in regulating pancreatic β cell function and the development of T2D.

## Methods

**Ethics statement**. For animal studies, this study complies with all relevant ethical regulations related to the use of research animals and was approved by the Animal Care and Use Committee of Shandong Eye Institute, Shandong First Medical University & Shandong Academy of Medical Sciences. For human studies, this study complies with all relevant ethical regulations for work with human participant and was approved by the Human Organ Donation Ethics Committee of the affiliated hospital of Qingdao University. Informed consent was obtained from the immediate family member of the donor. No study participant received compensation.

**Mice**. Insulin-2 promoter Cre mice (Ins2-Cre, Cat#003573) and miR-21$^{flox/flox}$ mice (Cat#NM-CKO-00051) were on the C57BL/6 background and purchased from Shanghai Biomodel Organism Science & Technology Development, Co., Ltd, China. C57BL/6 mice (Cat#000664) were purchased from SPF (Beijing) Biotechnology Co., Ltd, China. Mice with pancreatic β cell specific deletion of miR-21(*miR-21βKO*) were generated by crossing Ins2-Cre mice with miR-21$^{flox/flox}$ mice. ob/ob mice (Cat#T001461) were purchased from Jiangsu GemPharmatech, China. All animals were on the C57BL/6 background except db/db mice (BKS.Cg-Dock7$^m$ +/+ Lepr$^{db}$/J, Cat#000642), which were on the BKS background and purchased from Changzhou Cavens laboratory animals Co., Ltd, China. All mice were kept under pathogen-free conditions at the animal core facility of the Shandong Eye Institute. Mice were fed with sterile food (Cat#2212, Beijing Keao Xieli Feed Co., Ltd.) and autoclaved water without antibiotics and housed in a controlled environment (12-h light/dark cycle; humidity, 50–60%; ambient temperature, 22 ± 2 °C).

**Pancreas donor**. Human pancreas was obtained from a healthy donor (male, 50–60 years old) who was died of brain death caused by heavy cerebral trauma.

**Intraperitoneal glucose tolerance testing (IPGTT)**. Mice were fasted for 4 h and then intraperitoneally injected with 2 g/kg body weight D-glucose in PBS. Alternatively, high-fat diet-fed mice were fasted for 16 h and then intraperitoneally injected with 1 g/kg body weight D-glucose (Cat#G7528, Sigma) in PBS. Blood glucose level was measured using glucometer (Cat#HEA-232, OMRON).

**Islets isolation**. Islets from murine pancreata were isolated as described in our earlier publication[34]. In brief, a total of 3 mL collagenase P solution (0.4 mg/mL, Cat#11213873001, Roche) was slowly injected into the common bile duct after occlusion of the distal end proximal to the duodenum. The distended pancreas was excised and digested with collagenase P at 37 °C for 15 min. The collagenase digest was then subjected to density gradient centrifugation in Histopaque-1077 (Cat#10771, Sigma) to facilitate the harvest of islets. Islets were finally hand-picked using a micropipette under stereomicroscope. For human islets isolation, pancreas was stored and transported in organ perfusion fluid. After removal of all extraneous connective tissue and fat, pancreas was hand distended for 5 min with 2 mg/ml collagenase P solution supplemented with 20 mg/ml bovine serum albumin (BSA, Cat#A1933, Sigma) and 2 mg/ml soybean trypsin inhibitor (Cat#T9003, Sigma). Following the complete distension of the pancreas with enzyme solution, the organ was chopped into pieces and digested at 37 °C for 20 min. The collagenase digest was then subjected to Ficoll (Cat#17144002, Cytiva) density gradient separation. Islets were finally hand-picked using a micropipette under stereomicroscope.

**High glucose and high fat treatment**. Pancreatic islets were treated with either low glucose (5.5 mM)+ no fat, high glucose (25 mM)+no fat, low glucose (5.5 mM)+ sodium palmitate (0.4 mM, Cat#KT002, Kunchuang biotechnology) or high glucose (5.5 mM)+ sodium palmitate (0.4 mM). After 24 h, islets were collected and used for quantitative RT-PCR analysis.

**Insulin release test**. For IPGTT, serum was obtained from tail vein and insulin level was measured using a mouse insulin ELISA kit (Cat#EMINS, Thermo Fisher Scientific); For glucose-stimulated insulin secretion test (GSIS) in vitro, ~20-30 islets of 150–200 µm diameter were kept in 48-well plate and pre-incubated for 1 h at 37 °C in Krebs–Ringer bicarbonate (KRB) buffer (Cat#AAPR108, PYTHONBIO) containing 0.5% bovine serum albumin and 2.8 mM glucose. The buffer was then replaced with KRB buffer supplemented with 2.8 mM or 16.7 mM glucose. After incubation for 1 h at 37 °C, islets were pelleted by a 3 min centrifugation at 800 g, and media were aspirated and frozen for insulin determination. Insulin was also acid-ethanol extracted (1 mol/L HCl/70% ethanol) at −20 °C for 24 h from the islets and cellular debris removed by centrifugation at 15,000 g for 20 min. Insulin levels in the media and islet extract were determined as mentioned above. Total islet insulin content was defined as insulin level in the media plus islet extract. Insulin release was shown as the insulin level in the media as well as the fraction released of the total islet insulin content. Glucose-stimulated insulin release by human islets was determined using a human insulin ELISA kit (Cat#ab100578, Abcam).

**Histological and immunofluorescence analysis**. To obtain histological profiles of pancreas, pancreatic tissues were fixed in 10% formalin, embedded in paraffin, sectioned, stained with hematoxylin and eosin, and examined by microscopy. For immunofluorescence staining, pancreatic tissues were fixed in 4% paraformaldehyde (Cat#G1101, Servicebio), OCT (Cat#4583, Tissue-Tek) embedded, and sectioned. Sections were permeabilized in PBS containing 0.3% Triton X-100 (Cat#T8787, Sigma) and blocked in QuickBlock™ Blocking Buffer (Cat#P0260, Beyotime) prior to incubation with primary antibody against insulin (Cat#4590 S, CST, 1:100 dilution in blocking buffer) or Glut2 (Cat#sc-518022, C-10, Santa Cruz, 1:200 dilution in blocking buffer). Sections were then washed and incubated with Alexa Fluor 594-conjugated Donkey anti-Rabbit IgG (H + L) (Cat#711-585-152, Jackson, 1:200 dilution in blocking buffer) or Alexa Fluor 488-conjugated Donkey anti-Mouse IgG (H + L) (Cat#715-545-150, Jackson, 1:200 dilution in blocking buffer). After washing, sections were incubated with DAPI (Cat#H-1200, Vectorlabs) to stain the nuclear. Images were acquired on a confocal laser scanning microscope using Zen software (v2.3, Carl Zeiss) and analyzed using Image pro software (v6.0, Media Cybernetics).

**In vivo pancreas transfection**. On day 0, 3, 6, and 9, control (Cat#miR4N0000002-4-5, RiboBio) or miR-21 agomir (Cat#miR40004628-4-5, 5′-UAGCUUAUCAGACUGAUGUUGA-3′, RiboBio) were administrated through intravenous injection to 6-week old male type 2 diabetic db/db mice using pancreas in vivo transfection reagent (Cat#5051, Altogen Biosystems). Body weight and 5 h fasting blood glucose level were measured on day −1, 4, 7, and 10.

**Measurement of metabolic parameters**. Mice were fed with high-fat diet (Cat# D12451, Research DIETS) for 27 weeks. The serum levels of triglycerides (TG), total cholesterol (TC), high-density lipoprotein cholesterol (HDLC), and low-density lipoprotein cholesterol (LDLC) were measured using commercially available kit (Cat#A110-1-1, Cat#A111-1-1, Cat#A112-1-1, and Cat#A113-1-1, Nanjing Jiancheng Bioengineering Institute).

**G6P assay**. Islets treated with 2.8 mM or 16.7 mM glucose were lysed and G6P level in the lysate was determined using G6P Assay Kit with WST-8 (Cat#S0185, Beyotime).

**RNA sequencing**. Islets from WT and *miR-21βKO* mice were cultured at ~50 islets/well on 48-well plate and pre-incubated for 1 h at 37 °C in Krebs–Ringer bicarbonate (KRB) buffer containing 0.5% bovine serum albumin and 2.8 mM glucose. The buffer was then replaced with KRB buffer supplemented with

16.7 mM glucose, and incubated for another 20 min. Total RNA was prepared using TRIzol Reagent (Cat#15596018, Invitrogen). DNase I was used to digest DNA in total RNA and rRNA was removed by hybridization of complementary DNA oligos to rRNA followed by degradation of the RNA:DNA hybrids using RNase H (Cat#18021014, Thermo Fisher Scientific). Purified mRNA from previous steps was fragmented into small pieces and then used to prepare mRNA sequencing library. For quality control, libraries were validated using Agilent Technologies 2100 bio-analyzer (Agilent Technologies). The final library was amplified via rolling circle amplification with phi29 DNA polymerase (Cat#EP0092, Thermo Fisher Scientific), which could result in DNA nanoballs (DNB) that had more than 300 copies of one original molecule. DNBs were then loaded into the patterned nanoarray and pair end 150 bases reads were generated on a BGISEQ500 platform (BGI Shenzhen). The fold change was calculated from the normalized expression values, and statistical significance was calculated using the DEGseq2 software (v1.4.5, Bioconductor). A fold change greater than or equal to 1.5 and a $P$-value ≤ 0.05 was defined as significant. GO enrichment analysis of annotated different expressed gene was performed by cluster Profiler.

**Oxygen consumption rate (OCR) test**. OCR was measured using a Seahorse XFp analyzer (v1.1.1.3, Agilent Technologies) according to the company's recommendations. Briefly, XFp sensor cartridge was first hydrated in sterile water at 37 °C in a non-CO2 incubator. After 12 h, the cartridge was further hydrated in XF Calibrant (Cat#103059-000, Agilent Technologies) at 37 °C in a non-CO2 incubator for 1 h. Seahorse XFp cell culture plate was coated with 100 mg/mL poly-D-lysine (Cat# P0899, Sigma) for 1 h at room temperature. Excess poly-D-lysine was removed, and culture plate was dried for 30 min at 37 °C before it is ready for further experiment. ~30 islets from WT and *miR-21βKO* mice were resuspended in Seahorse glucose-free DMEM (Cat#103575-100, Agilent Technologies) supplemented with 2 mM glutamine (Cat#103579-100, Agilent Technologies). After islets were seeded on coated XFp culture plate, the plate was centrifuged for 5 min at 500 $g$ with brake and then pre-incubated for 1 h at 37 °C without CO2. After Seahorse XFp Analyzer was calibrated with hydrated cartridge, the XFp culture plate was transferred into the analyzer and OCR was measured. Basal OCR was recorded in base medium for 6 measurement cycles (3-min mix, 0-min wait, and 3-min measure) before injecting 16.7 mM D-glucose (Cat#103577-100, Agilent Technologies) to stimulate cellular oxygen consumption. The measured OCR was analyzed using the Seahorse Wave software (v2.6.1, Agilent Technologies).

**Glucose uptake**. Glucose uptake was performed according to manufacturers' instructions (Cat# 600470, Cayman chemical). Briefly, ~30 islets from WT or *miR-21βKO* mice were seeded in 96-well plate in 100 μl glucose-free culture medium. Fluorescence-labeled 2-deoxy-glucose analog (2-NBDG) was added to a final concentration of 150 μg/ml. After incubation at 37 °C for 10 min, islets were washed and dissociated into single cells using trypsin digestion. Cells were then washed, resuspended in 200 μl Cell-Based Assay Buffer, and analyzed immediately using CytoFLEX (Beckman Coulter Inc). Data were collected using CytExpect (v2.1, Beckman Coulter Inc) and analyzed using FLOWJo (v10, FlowJo). Glucose uptake was determined by calculating the mean fluorescence intensity (MFI) of 2-NBDG.

**RNA isolation and qRT-PCR**. Total RNA was extracted from mouse islets or β-TC-6 cells using Trizol reagent. cDNA was generated using a reverse transcription kit with gDNA Eraser (Cat#RR047A, Takara). Real-time PCR was carried out in a CFX96 real-time system (BIO-RAD, USA) using SYBR Green PCR master mix (Cat#QPS-201(-), TOYOBO). The primer sequences for the detection of mouse Gluts, Pdcd4 and insulin are as follows:
mGlut1-RT-F: 5′-TGTCCTATCTGAGCATCGTG-3′; mGlut1-R: 5′-CTCCT CGGGTGTCTTATCAC-3′; mGlut2-F: 5′-CTGCTCTTCTGTCCAGAAAGC-3′; mGlut2-R: 5′-TGGTGACATCCTCAGTTCCTC-3′; mGlut3-F: 5′-TCAACC GCTTTGGCAGACGCA-3′; mGlut3-R: 5′-AGGCGGCCCAGGATCAGCAT-3′; mGlut4-F: 5′-CTGCAAAGCGTAGGTACCAA-3′; mGlut4-R: 5′-CCTCCCGCCC TTAGTTG-3′; Glut5-F: 5′-CCAATATGGGTACAACGTAGCTG-3′; mGlut5-R: 5′-GCGTCAAGGTGAAGGACTCAATA-3′; mPdcd4-F: 5′-ATGGATATAG AAAATGAGCAGAC-3′; mPdcd4-R: 5′-CCAGATCTGGACCGCCTATC-3′; mInsulin1-F: 5′-CACCTGGAGACCTTAATGGGCC-3′; mInsulin1-R: 5′-GGT AGAGAGCCTCTACCAGGTG-3′. The primer sequences for the detection of human Glut2 are as follows: hGLUT2-F: 5′-GCCTGGTTCCTATGTATATCGG T-3′; hGLUT2-R: 5′-GCCACAGATCATAATTGCCCAAG-3′. Relative levels of gene expression were determined using mouse or human GAPDH as the internal control (mGapdh-F: 5′-GGTCGGTGTGAACGGATTTGG-3′; mGapdh-R: 5′-CCG TGAGTGGAGTCATACTGGAA-3′; hGAPDH-F: 5′-AGATCCCTCCAAAATCA AGTGG-3′; hGAPDH-R: 5′-GGCAGAGATGATGACCCTTTT-3′).

For the detection of miR-21 level, reverse transcription and quantitative PCR were performed using specific primers for miR-21 (Cat#mqpscm001-5, RiboBio) and control U6 (Cat#MQPS0000002-1-100, RiboBio).

**Ago2 immunoprecipitation**. Ago2 immunoprecipitation was performed according to the protocol provided with the Magna RIP RNA-Binding Protein Immunoprecipitation Kit (Cat#17-701, Merck Millipore). Briefly, $1 \times 10^7$ MIN6 cells (Cat#CL-0674, Procell) were transfected with control (Cat#miR3N0000002-4-5,

RiboBio) or miR-21 antagomir (Cat#miR3CM001, 5′-UCAACAUCAGUCU-GAUAAGCUA-3′, RiboBio) using Lipofectamine™ RNAiMAX (Cat#13778-150, Invitrogen). Cells were then collected by centrifugation at 500 $g$ for 5 min and then lysed with RIP lysis buffer. For the magnetic bead/Ab preparation, 5 μg of anti-Ago2 (Cat#03-110, Merck Millipore) or normal mouse IgG were incubated with beads for 30 min at room temperature. Supernatant from the RIP lysate was obtained by centrifugation at 14,000 rpm for 10 min and incubated overnight at 4 °C with bead-Ab complex. 10 μl cell lysate was used as input for normalization purpose. Immunoprecipitated RNAs were purified and analyzed by quantitative RT-PCR for the following mouse genes: mPdcd4-F: 5′-ATGGATATA-GAAAATGAGCAGAC-3′; mPdcd4-R: 5′-CCAGATCTGGACCGCCTATC-3′; mPten-F: 5′-TGGATTCGACTTAGACTTGACCT-3′; mPten-R: 5′-GCG GTGTGTCATAATGTCTCTCAG-3′; mPtger3-F: 5′-CCGGAGCACTCTGCTG AAG-3′; mPtger3-R: 5′-CCCCACTAAGTCGGTGAGC-3′; mAdtrp-F: 5′-CTTT CCAGCCTGAACAGAGG-3′; mAdtrp-R: 5′-TTCCAATCTGTGGGATGTGA-3′; mEgf-F: 5′-CGGATGGTACGAATGGTGCAG-3′; mEgf-R: 5′-GTACCTTCT GTCTACTCCCAG-3′; mLrrc55-F: 5′-CCTCTAACACAGTCCACTCAG-3′; mLrrc55-R: 5′-TAGCCCTTGACCAGCATAAGC-3′; mAcvr1-F: 5′-CCCAAG AGTAACAAGAGTAAC-3′; mAcvr1-R: 5′- GCCTAGATACTACTGTTCTGC-3′.

**Adenovirus infection**. Adenovirus over-expressing mouse Glut2, mouse miR-21 and human miR-21 under the control of CMV promoter or mouse insulin-1 promoter were designed and supplied by WZ Biosciences Inc., China. Islets were cultured in complete DMEM medium (containing 5.5 mM glucose) at ~30 islets/well on 96-well plate and $1 \times 10^8$ pfu virus was added to the plate. After 12 h, replace culture medium with fresh complete DMEM medium and continue to culture the islets for 24–48 h. Transduction efficiency was determined using confocal microscope.

**Apoptosis detection**. Fresh islets or islets infected with virus were dissociated into single cells using trypsin digestion (Cat#T4674, Sigma) and apoptosis was examined using an apoptosis detection kit (Cat#KGA1026, KeyGen BioTECH). Briefly, cells were stained with Annexin-V-APC and 7AAD followed by analysis using CytoFLEX. Data were collected using CytExpect and analyzed using FLOWJo.

**Transient transfection**. For siRNA transfection, β–TC-6 cells (Cat#CRL-11506, ATCC) were transfected using Lipofectamine 3000 transfection reagent (Cat#, L3000-015, Invitrogen). A universal negative control siRNA duplex as well as siRNA duplex targeting mouse Pdcd4 (5′-GGAAGUGAAGCGGUUAGAATT-3′) were designed and synthesized by RiboBio, China. After 72 h, cells were harvested and used for western blot analysis. For luciferase assay, β–TC-6 cells were transiently transfected with mouse *Glut2* promoter luciferase reporter (−350 to +97) together with the pRK5-c-Jun or empty vector using Lipofectamine 3000 transfection reagent. After 36 h, the luciferase activities of total cell lysates were measured using the Dual-Luciferase Reporter Assay system (Cat#E1910, Promega). Co-transfection of the Renilla luciferase expression vector pRL-TK (Cat#E2241, Promega) was used as internal control. For the detection of Glut2 mRNA expression, β–TC-6 cells were transfected with c-Jun expression vector (pRK5-c-Jun) or empty vector (pRK5). After 24 h, cells were harvested and used for quantitative RT-PCR.

**AP-1 inhibition**. For the detection of Glut2 mRNA expression under the condition of AP-1 inhibition, β–TC-6 cells were treated with DMSO or AP-1 inhibitor T-5224 (80 μM, Cat# HY-12270, MedChem Express). After 12 h, cells were harvested and used for quantitative RT-PCR analysis. For the detection of Glut2 protein expression under the condition of AP-1 inhibition, pancreatic islets were isolated from C57BL/6 mice and treated with DMSO or AP-1 inhibitor T-5224 (80 μM). After 24 h, islets were collected and used for western blot analysis.

**Western blotting**. Islets or β-TC-6 cells were lysed using RIPA lysis buffer containing protease inhibitors (Cat#1861280, Thermo Fisher Scientific). Protein extracts were separated by 10% SDS-PAGE gel and transferred to the PVDF membrane (Cat#IPVH00010, Merck Millipore). After blocking with 5% skim milk, membrane was incubated overnight with one of the following primary antibodies: Glut2 (Cat#NBP2-22218, Novus, 1:2000 dilution in blocking buffer), Pdcd4 (Cat#9535 T, D29C6, CST, 1:1000 dilution in blocking buffer), c-Jun (Cat#9165T, 60A8, CST, 1:1000 dilution in blocking buffer), β-actin (Cat#sc-69879, AC-15, Santa Cruz, 1:500 dilution in blocking buffer), or Gapdh (Cat#5174, D16H11, CST, 1:1000 dilution in blocking buffer). Membranes were then incubated for 1 h at room temperature with goat anti-mouse-HRP antibody (Cat#SA00001-1, Proteintech, 1:5000 dilution in blocking buffer) or goat anti-rabbit-HRP antibody (Cat#SA00001-2, Proteintech, 1:5000 dilution in blocking buffer). Signals were detected by chemiluminescence assay using ChemiDoc Touch (Bio-Rad) and Image Lab Touch software (v1.2.0.12, Bio-Rad). Uncropped scans of all blots are supplied as Source Data files.

**Identification of putative miR-21 binding sites**. TargetScan (v7.2, Whitehead Institute) was used to search for putative miR-21 binding sites in the 3′-UTR of genes that were up-regulated in the islets from *miR-21βKO* mice and also involved in the regulation of insulin secretion.

**Statistical analysis**. All statistical analyses were performed using GraphPad Prism (v8.0, GraphPad Software). The significance of the differences between two groups in blood glucose, insulin concentration, β cell counts, and RNA expression level were determined by unpaired Student's $t$ test (two-sided). One-way ANOVA was utilized for multi-group comparison, followed by Bonferroni post hoc analysis. For glucose tolerance test, glucose uptake test and comparisons of body weight and blood glucose level of mice fed with high-fat diet, statistical significance was analyzed using two-way ANOVA with Bonferroni correction. $P < 0.05$ was considered to be significant.

**Reporting summary**. Further information on research design is available in the Nature Research Reporting Summary linked to this article.

## Data availability

The RNAseq data that support the findings of this study have been deposited in Gene Expression Omnibus (GEO) with the accession code GSE191194. The authors declare that all other data supporting the findings of this study are available within the paper and its supplementary information files. Source data are provided with this paper.

## Code availability

No custom code or mathematical algorithm is deemed central to the conclusions of this paper.

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

## Acknowledgements

This study was supported by Shenzhen Science and Technology Innovation Commission (JCYJ20170413165432016, Q.R), Qingdao Municipal Science and Technology Bureau (21-1-4-rkjk-11-nsh, Q.R), Natural Science Foundation of Shandong Province (ZR2021QH013, R.L), and Postdoctoral Innovation Project of Shandong Province (202103021, R.L).

## Author contributions

Q.R. designed research; C.L., R.L., X.H., and H.Y. performed research; R.L. and P.S. contributed reagents/analytic tools; C.L., R.L., B.Z., W.S., and Q.R. analyzed data; R.L., C.L., and Q.R. wrote the paper.

## Competing interests

The authors declare no competing interest.
