## [Peer Review File · Nature Communications]

Title: MicroRNA-21 Promotes Pancreatic β cell Function through Modulating Glucose UptakeREVIEWER COMMENTS

Reviewer #2 (Remarks to the Author):

In this manuscript, Liu et al. studied the role of miR-21 in pancreatic beta cell function. By using a beta cell-specific knockout mouse model, they discovered that miR-21 deletion results in glucose intolerance in mice due to impaired glucose-stimulated insulin secretion. Mechanistically, their studies suggested that miR-21 deletion leads to decreased glucose uptake by reducing the glucose transporter Glut2 in a miR-21-Pdcd4-AP-1 dependent pathway. They also found that delivery of miR-21 to db/db mice promoted Glut2 expression in islets, increased serum insulin levels, and decreased blood glucose levels. They therefore propose that miR-21 in islet beta cells promotes insulin secretion.

Overall, the manuscript is well written, includes a large body of work and represents the first time that a beta cell specific knockout mouse model was used to investigate the function of miR-21 in beta cells. On the other hand, the role of miR-21 in beta cell apoptosis and function and the connection of miR-21 and GLUT2 have been studied previously, and in fact, the current data conflicts with some of the previous work (e.g. Roggli et al.) as acknowledged by the authors. In addition, the manuscript raises a number of issues that would need to be addressed:

1. Results from the miR-21 β KO mice are at odds not only with previous in vitro studies demonstrating the opposite effect (Roggli et al.), but more importantly also with the authors own current findings that miR21 was increased in response to glucolipototoxicity and in db/db islets. The authors are trying to explain this as a “negative feedback” or compensatory attempt to improve insulin secretion and hyperglycemia, but if so, one would expect that in models of compensated insulin resistance such as in the ob/ob mice, miR21 expression should be even higher, which it was not. Furthermore, miR21 was also found to be increased in human islets of insulin resistant subjects.
2. It would be helpful if the authors could validate some findings in human islets. For example, use ade-miR-21 to transfect human islets and see if GSIS and GLUT2 are increased.
3. Since the authors introduced that the role of miR-21 in apoptosis has been well studied in other tissues and beta cells, they should address why there is no difference in beta cell apoptosis/survival in islets of miR-21 β KO mice and Ade-miR-21 treated mice.
4. In any case, the suggestion of miR21 increase as a therapeutic approach is highly problematic given the oncogenic potential of miR21 and the fact that its presumed target in this case is PDCD4, a known tumor suppressor gene.
5. Finally, there is a large body of literature about miR21 exosomes that should also be taken into account.

Reviewer #4 (Remarks to the Author):

Liu and colleagues investigated the role of miR-21 in mouse pancreatic beta cells using mice with beta-cell specific deletion of miR-21 (miR-21 β KO). They found that these mice developed glucose intolerance due to impaired glucose-stimulated insulin secretion (GSIS) via downregulation of Glut2 expression.

The authors have clear aims and the study design and methods employed were appropriate to support some but not all of the stated conclusions. Many of the experiments performed adequately supported/recapitulated previous knowledge in the field. However the main claim that miR-21 is involved in Glut2 regulation is not sufficiently supported by the findings presented in this study.

Major comments:

1. In the introduction, the authors mentioned two references showing that increased miR-21 in islets resulted in (a) glucose intolerance in humans (b) reduced GSIS in MIN6 cells. How do you reconcile these findings which are completely opposite with what you observed in the miR-21 β KO mice? To address this discrepancy, the appropriate model should be a transgenic mouse over-expressing miR-21 in the beta cell.

2. The db/db mice also showed increased miR-21 expression. However it has also been shown in earlier studies that Glut2 is downregulated in this mouse model. This contradicts your finding that miR-21 can actually reduce Glut2 expression. It is apparent that different regulatory mechanisms are happening in different animal models with regards to miR-21 dependent regulation of Glut2. Given the close to 900 genes that are significantly deregulated in the miR21 β KO model, it is highly unlikely that the regulation of Glut2 will only be dependent on the indirect effect of miR-21 on Pdc4.

3. MiRNA-mediated regulation of biological pathways is highly dependent on direct negative transcriptional regulation of the targets in that specific pathway. Despite the apparent richness of data generated from this study (eg RNA-seq of WT vs miR21 β KO islets), no direct miR-21 target was established in this study that could possibly link to Glut2 regulation. The reduction of c-Jun in the miR21 β KO islet is circumstantial at best. A biochemical assay such as Ago-RIP to establish direct miRNA-target interaction should be performed.

4. It is very important to deposit sequencing data in any public repository to allow for data validation.

Minor comments:

1. In line 152, it was mentioned that "rRNA was removed by RNase H digestion." Wouldn't this also result to all RNAs in the prep getting digested?

2. Line 157: 300 copies of one molecular. ??

3. In methods, line 192. Where's the description of qPCR for miR-21? Which miRNA was used as endogenous control?

Response to Reviewers

We thank the reviewers and editor for the constructive comments and insightful analyses of this manuscript. In this revised manuscript, we have tried to address all the issues raised thoroughly. As a consequence, the quality of this work has been significantly improved. We provide below our point-by-point response (shown in blue fonts) to comments from the reviewers.

Reviewer #2 (Remarks to the Author):

In this manuscript, Liu et al. studied the role of miR-21 in pancreatic beta cell function. By using a beta cell-specific knockout mouse model, they discovered that miR-21 deletion results in glucose intolerance in mice due to impaired glucose-stimulated insulin secretion. Mechanistically, their studies suggested that miR-21 deletion leads to decreased glucose uptake by reducing the glucose transporter Glut2 in a miR-21-Pdcd4-AP-1 dependent pathway. They also found that delivery of miR-21 to db/db mice promoted Glut2 expression in islets, increased serum insulin levels, and decreased blood glucose levels. They therefore propose that miR-21 in islet beta cells promotes insulin secretion.

Overall, the manuscript is well written, includes a large body of work and represents the first time that a beta cell specific knockout mouse model was used to investigate the function of miR-21 in beta cells. On the other hand, the role of miR-21 in beta cell apoptosis and function and the connection of miR-21 and GLUT2 have been studied previously, and in fact, the current data conflicts with some of the previous work (e.g. Roggli et al.) as acknowledged by the authors. In addition, the manuscript raises a number of issues that would need to be addressed:

1. Results from the miR-21 β KO mice are at odds not only with previous in vitro studies demonstrating the opposite effect (Roggli et al.), but more importantly also with the authors own current findings that miR21 was increased in response to glucolipototoxicity and in db/db islets. The authors are trying to explain this as a “negative feedback” or compensatory attempt to improve insulin secretion and hyperglycemia, but if so, one would expect that in models of compensated insulin resistance such as in the ob/ob mice, miR21 expression should be even higher, which it was not. Furthermore, miR21 was also found to be increased in human islets of insulin resistant subjects.

Response:

1) Regarding why the results from *miR-21 β KO* mice are contrary to previous report, we have discussed this in the manuscript and speculate that it maybe be related to different cell types used (mouse cell line vs primary mouse islets) or difference in the manipulation of miR-21 expression (over-expression vs genetic deletion), although this hypothesis remains to be further clarified. Nevertheless, we have shown that delivery of miR-21 into the pancreas of type 2 diabetic mice significantly reduced blood glucose level.

- 2) Our results showed that mouse islets treated with combined high glucose and high fat but not high glucose or high fat alone significantly up-regulated miR-21 expression (Fig. 1C). These are indeed consistent with the data that miR-21 expression in the islets from *db/db* mice (display both severe obesity and severe hyperglycemia) but not *ob/ob* mice (display severe obesity but mild hyperglycemia) was significantly increased when compared to control mice.
- 3) It is true that *ob/ob* mice were insulin resistant and produce significant more insulin when compared to control mice. The increase in insulin production was probably due to other mechanisms not related to miR-21, since miR-21 expression in the islets from *ob/ob* mice was not increased. When mice were both insulin resistant and hyperglycemia (for example, *db/db* mice), islet miR-21 expression will increase and promotes insulin secretion in a “negative feedback” manner.
- 4) It has been reported that miR-21 was increased in glucose intolerant patients. Since authors consider the donor with an HbA1c ≥ 6.1 as glucose intolerant, it is not known whether these patients were insulin resistant subjects (with or without obesity) or suffered from type 1 diabetes. In one hand, it has been reported that miR-21 expression is increased in the islets of type 1 diabetic patients. On the other hand, since our assumption is that islet miR-21 expression will be up-regulated only under the condition of combined high glucose and high fat, we speculate that islet miR-21 expression will only increase in patients with both insulin resistance and obesity, but not in patients with insulin resistance only.

2. It would be helpful if the authors could validate some findings in human islets. For example, use ade-miR-21 to transfect human islets and see if GSIS and GLUT2 are increased.

Response:

We have used ade-miR-21 to transduce human islets and examined GLUT2 expression and GSIS. Our results showed that over-expression of miR-21 in human islets can also increase GLUT2 expression and promote GSIS. These new data have been added to Figure 7 (C & D).

3. Since the authors introduced that the role of miR-21 in apoptosis has been well studied in other tissues and beta cells, they should address why there is no difference in beta cell apoptosis/survival in islets of miR-21 β KO mice and Ade-miR-21 treated mice.

Response:

Although the role of miR-21 in the beta-cell apoptosis has been extensively studied, there is still no consensus on the exact role miR-21 plays. One group found that increasing miR-21 expression in INS1 cells promotes the apoptosis induced by inflammatory cytokines [1]. Another group reported that inhibiting the expression of miR-21 in MIN6 cells does not affect the apoptosis induced by inflammatory cytokines [2]. Our previous studies have found that inhibiting miR-21 expression in β -TC-6 cells promotes the apoptosis and *miR-21* β KO islets are more sensitive to apoptosis induced by inflammatory cytokines (we didn't include these data in the current manuscript because they are the focus of another project of ours which

studies the role of miR-21 in islet beta cells during the development of type 1 diabetes). In our current study, we found that there is no significant difference in apoptosis between control and miR-21-overexpressed islets. We believe that the above-mentioned inconsistency may be related to the following factors: 1) apoptosis-inducing conditions; 2) the way miR-21 expression was altered (overexpression or inhibition); 3) types of islet β -cell (cell line or primary islets); 4) the duration of altered miR-21 expression (transient or stable). We have discussed these in the revised manuscript (line 609-622).

[1] Sims EK, Lakhter AJ, Anderson-Baucum E, Kono T, Tong X, Evans-Molina C. MicroRNA 21 targets BCL2 mRNA to increase apoptosis in rat and human beta cells. *Diabetologia*. 2017;60(6):1057-1065.

[2] Roggli E, Britan A, Gattesco S, et al. Involvement of microRNAs in the cytotoxic effects exerted by proinflammatory cytokines on pancreatic beta-cells. *Diabetes*. 2010;59(4):978-986.

4. In any case, the suggestion of miR21 increase as a therapeutic approach is highly problematic given the oncogenic potential of miR21 and the fact that its presumed target in this case is PDCD4, a known tumor suppressor gene.

Response:

MiR-21 has been reported to be overexpressed in pancreatic cancer and miR-21 inhibition has been used for pancreatic cancer therapy. However, although overexpression of miR-21 in pancreatic cancer cell line increased cell growth, it is unlikely that overexpression of miR-21 alone in normal islet β -cell will lead to tumorigenesis, since one group reported that globally over-expressing miR-21 in mice will not initiate lung cancer development [3] and another group reported that no pancreatic cancer was observed after overexpressing miR-21 in mice [4]. These data suggest that miR-21 overexpression alone in normal islet β -cell is not sufficient for islet β -cell tumorigenesis, though various strategies (for example, transient expression and islet β -cell specific expression) are still needed to minimize the side effects when increasing miR-21 in the islets was adopted as a therapeutic approach in the future. On the other hand, one study surprisingly revealed an overall tumor suppressive function of miR-21 in *in-vivo* pancreatic cancer adenocarcinoma model [5]. Therefore, future studies will be needed to evaluate the therapeutic potential of modulation of miR-21 activity. We have rephrased the sentences in the revised manuscript as regarding to therapeutic potential of modulating miR-21 level in the pancreas (line 593-594).

[3] Hatley ME, Patrick DM, Garcia MR, et al. Modulation of K-Ras-dependent lung tumorigenesis by MicroRNA-21. *Cancer Cell*. 2010;18(3):282-293.

[4] Medina PP, Nolde M, Slack FJ. OncomiR addiction in an *in vivo* model of microRNA-21-induced pre-B-cell lymphoma. *Nature*. 2010;467(7311):86-90.

[5] Schipper J, Westerhuis JJ, Beddows I, et al. Loss of microRNA-21 leads to profound stromal remodeling and short survival in K-Ras-driven mouse models of pancreatic cancer. *Int J Cancer*. 2020;147(8):2265-2278.

5. Finally, there is a large body of literature about miR21 exosomes that should also be taken into account.

Response:

We thank the reviewer for this excellent suggestion. It has been reported that miRNAs in islet β -cell-derived exosome play an important role in insulin resistance and type 2 diabetes. Indeed miR-21 is enriched in islet β -cell-derived exosome. Although we have shown that miR-21 in islet β -cell promotes insulin secretion, it is possible that exosomal miR-21 released by islet β -cell may regulates the development of diabetes. We have discussed this in the revised manuscript (line 570-575). On the other hand, we currently have an ongoing project focusing on the role of islet exosomal miR-21 during the development of type 1 diabetes.

Reviewer #4 (Remarks to the Author):

Liu and colleagues investigated the role of miR-21 in mouse pancreatic beta cells using mice with beta-cell specific deletion of miR-21 (miR-21 β KO). They found that these mice developed glucose intolerance due to impaired glucose-stimulated insulin secretion (GSIS) via downregulation of Glut2 expression.

The authors have clear aims and the study design and methods employed were appropriate to support some but not all of the stated conclusions. Many of the experiments performed adequately supported/recapitulated previous knowledge in the field. However the main claim that miR-21 is involved in Glut2 regulation is not sufficiently supported by the findings presented in this study.

Major comments:

1. In the introduction, the authors mentioned two references showing that increased miR-21 in islets resulted in (a) glucose intolerance in humans (b) reduced GSIS in MIN6 cells. How do you reconcile these findings which are completely opposite with what you observed in the miR-21 β KO mice? To address this discrepancy, the appropriate model should be a transgenic mouse over-expressing miR-21 in the beta cell.

Response:

- 1) According to published reference, miR-21 was found to be increased in the islets of glucose intolerant patients. However, it is not known whether increased miR-21 in islets resulted in glucose intolerance or improved glucose tolerance. Our current study suggests that miR-21 in islets promotes insulin secretion in a “negative feedback” manner.
- 2) Regarding why the results from *miR-21* β KO mice are contrary to previous report (reduced GSIS in MIN6 cells), we have discussed this in the manuscript (line 543-546) and speculate that it maybe be related to different cell types used (mouse cell line vs primary mouse islets) or difference in the manipulation of miR-21 expression (transient over-expression vs genetic deletion), although this hypothesis remains to be further clarified. Nevertheless, we have shown both *in*

vitro and *in vivo* that miR-21 promotes GSIS in primary pancreatic islets. More importantly, we revealed the possible mechanism of miR-21 mediated promotion of GSIS and demonstrated that delivery of miR-21 into the pancreas of type 2 diabetic mice significantly reduced blood glucose level.

- 3) We agree with this reviewer's suggestion that transgenic mouse over-expressing miR-21 in islet β -cell will be the best option to address the discrepancy. However, since it may take too long to establish the transgenic mouse model, we decided to overexpress miR-21 specifically in primary mouse islet β -cell *in vitro* using an over-expression system under the control of mouse insulin-1 promoter. Our results showed that overexpression of miR-21 specifically in primary mouse islet β -cell promotes Glut2 expression and glucose stimulated insulin secretion. These new results have been added to Figure 7 (E & F).

2. The db/db mice also showed increased miR-21 expression. However it has also been shown in earlier studies that Glut2 is downregulated in this mouse model. This contradicts your finding that miR-21 can actually reduce Glut2 expression. It is apparent that different regulatory mechanisms are happening in different animal models with regards to miR-21 dependent regulation of Glut2. Given the close to 900 genes that are significantly deregulated in the miR21 β KO model, it is highly unlikely that the regulation of Glut2 will only be dependent on the indirect effect of miR-21 on Pcd4.

Response:

- 1) Although db/db mice exhibit both increased miR-21 expression and Glut2 down-regulation, it remains unclear whether miR-21 inhibits or promotes Glut2 expression. Our current study showed that overexpressing miR-21 in both human (new data) and mouse pancreatic islets promotes Glut2 expression. We further demonstrated that miR-21 indirectly promotes Glut2 expression through regulating Pcd4-AP-1-Glut2 pathway. The role of miR-21 up-regulation in diabetic db/db mice, as a result of high glucose/high fat, is to reduce glucose level through the promotion of Glut2 expression. To put it in another way, Glut2 level will be even lower if miR-21 was not up-regulated.
- 2) Although our transcriptomic analysis identified close to 900 genes dysregulated in the islets from miR-21 β KO mice, we found that only Pten (a well-known miR-21 target) and Adtrp (a potential miR-21 target identified by our Ago2-RIP analysis) displayed increased expression in the islets from miR-21 β KO mice and also may be involved in the regulation of insulin secretion determined by GO analysis. However, we can't exclude the possibility that miR-21 may regulate insulin secretion through targeting genes other than Glut2. Since our current study revealed that miR-21 promotes glucose uptake and doesn't enhance insulin secretion by an effect exerted distally at the level of insulin granule exocytosis, we think that the dysregulation of most genes may be secondary to the defect of glucose uptake. Nevertheless, we have discussed this in the revised manuscript (line 559-569).

3. MiRNA-mediated regulation of biological pathways is highly dependent on direct negative transcriptional regulation of the targets in that specific pathway. Despite the

apparent richness of data generated from this study (eg RNA-seq of WT vs miR21 β KO islets), no direct miR-21 target was established in this study that could possibly link to Glut2 regulation. The reduction of c-Jun in the miR21 β KO islet is circumstantial at best. A biochemical assay such as Ago-RIP to establish direct miRNA-target interaction should be performed.

Response:

- 1) Our current study indicates that miR-21 indirectly promotes Glut2 expression though regulating Pcdcd4-AP-1-Glut2 pathway. Pcdcd4 is a well-known functional target of miR-21 in most cancer cell lines. We further showed that Pcdcd4 is a direct target of miR-21 in β -cell through Ago2-RIP analysis (Figure S6). Therefore, miR-21 may promote Glut2 expression through direct targeting of Pcdcd4.
- 2) Our RNA-Seq analysis showed that 33 genes up-regulated in *miR-21* β KO islets were clustered into functional class that may be involved in insulin secretion. In order to determine which one of them may be the direct target of miR-21, we searched for potential miR-21 binding site using TargetScan. Genes were selected for further Ago2-RIP RT-PCR analysis when potential miR-21 binding site was identified in their 3'-UTR. Our results showed that, besides Pcdcd4 and Pten (two previously confirmed functional targets of miR-21), only Adtrp was identified to be a possible direct target of miR-21 (Fig. S6). Whether Adtrp negatively regulates insulin secretion in pancreatic islets and whether miR-21 promotes insulin secretion through the inhibition of genes other than Glut2 (such as Adtrp, Pten, etc) remains to be further studied. We have discussed this in the revised manuscript (line 559-569).

4. It is very important to deposit sequencing data in any public repository to allow for data validation.

Response:

We have deposited RNAseq data in "Gene Expression Omnibus" with the accession code GSE191194 (<https://www.ncbi.nlm.nih.gov/geo/query/acc.cgi?acc=GSE191194>). The secure token to access the data is "kfqbayymnjcphoh".

Accession code and associated hyperlink have also been provided in "Data Availability Statement".

Minor comments:

1. In line 152, it was mentioned that "rRNA was removed by RNase H digestion." Wouldn't this also result to all RNAs in the prep getting digested?

Response:

We are sorry for the confusion. Targeted depletion of rRNA was used to remove rRNA from total RNA. Specifically, complementary DNA oligos were used to hybridize to rRNA followed by degradation of the RNA:DNA hybrids using RNase H [1]. We have corrected this in the revised manuscript (line 164-167).

[1] Kraus AJ, Brink BG, Siegel TN. Efficient and specific oligo-based depletion of rRNA. *Sci Rep.* 2019; 9(1):12281.

2. Line 157: 300 copies of one molecular. ??

Response:

It should be “300 copies of one original molecule”. We have modified this sentence in the manuscript (line 169-172) as following: *The final library was amplified via rolling circle amplification with phi29 DNA polymerase (Thermo Fisher Scientific, MA, USA), which could result in DNA nano-balls (DNB) that had more than 300 copies of one original molecule.*

3. In methods, line 192. Where's the description of qPCR for miR-21? Which miRNA was used as endogenous control?

Response:

We are sorry for the lack of description of qPCR for miR-21 in the manuscript. For the detection of miR-21 level, reverse transcription and quantitative PCR were performed using specific primers for miR-21 and control U6 (RiboBio, China). We have added this description to revised manuscript (line 229-230).

REVIEWER COMMENTS

Reviewer #2 (Remarks to the Author):

While the authors have addressed some of the issues raised, significant concerns remain.

1) Additional studies in human islets seem to have been conducted, but there is no information about these islets, their donors (diabetic, obese, glucose intolerant?) or where they were obtained.

2) Under different conditions not shown in this manuscript, the authors apparently do see significant effects on apoptosis in response to modulation of miR-21. Since the shown results only demonstrate islet cell viability and don't address beta cell apoptosis, it would still be possible that the observed effects are due to an increase in insulin producing beta cells rather than increased insulin secretion per se. To address this issue, it would be helpful to at least also show islet insulin content and fractional insulin release.

3) The issue of therapeutic translatability has not been sufficiently toned down and there are still misleading statements raising questions about the overall understanding of the problems at hand. For example line 603-606 "Because of its high safety and effectiveness, adeno-associated viral (AAV) mediated delivery system has been widely used to treat various diseases, including diabetes. For example, AAV-mediated over-expression of glucokinase and insulin can significantly reduce blood glucose level in diabetic dog." is highly problematic as AAV mediated systems are not being used to treat human diabetes and are not appropriate for targeting pancreatic islets as they predominantly target the liver, which was probably the case for this unreferenced dog example.

Reviewer #5 (Remarks to the Author):

Comments:

Remarks to the Author

In this manuscript, Liu et al researched the role of miR-21 in mouse pancreatic beta cells using mice with beta-cell specific deletion of miR-21 (miR-21 β KO). They found that these mice developed glucose intolerance due to impaired glucose-stimulated insulin secretion (GSIS) via downregulation of Glut2 expression.

Major comments:

1. Author response it maybe be related to different cell types used (mouse cell line vs primary mouse islets) or difference in the manipulation of miR-21 expression. Thus, we think author at least research the function of miR-21a in the primary mouse islets.
2. Author response that they have overexpress miR-21 specifically in primary mouse islet -cell in vitro using an over-expression system under the control of mouse insulin-1 promoter. How to inject the over-expression system into the mouse. In the tail vein, even though the carrier contains insulin-1 promoter,

it cannot be specifically expressed in β cells? Author should state clearly in the manuscript.

3. As mentioned in the manuscript, RIP-cre mice were used for generation of β cell specific KO. It should be noted that the transgene in RIP-cre has been found to be expressed in the hypothalamus. Whether MiR-21 was different expression in the hypothalamus?

4. In the research, whether author tested off targets effects of miR-21 KO mice, please provide.

5. In the "Result 1", it is shown "We then speculate that islet miR-21 expression will be up-regulated only under the condition of combined high glucose and high fat. We confirmed this hypothesis by showing that mouse islets treated with combined high glucose and high fat but not high glucose or high fat alone significantly up-regulated miR-21 expression". It is not enough to draw the conclusion unless the author provides the expression of miR-21 in glucose and/or toxicity condition in vivo.

6. A part of the Western blots (Fig 6B, Fig 6D and Fig 7M and many more) are show incomplete. The author should provide unprocessed western blot images with markers as source data.

Response to Reviewers

We thank the reviewers for their constructive comments. We provide below our point-by-point response (shown in blue fonts) to comments from the reviewers.

Reviewer #2 (Remarks to the Author):

While the authors have addressed some of the issues raised, significant concerns remain.

1) Additional studies in human islets seem to have been conducted, but there is no information about these islets, their donors (diabetic, obese, glucose intolerant?) or where they were obtained.

Response:

In order to investigate whether over-expression of miR-21 in human islets was able to increase glucose stimulated insulin secretion, islets were obtained from the pancreas of a healthy donor (male, 56-year old) who was died of brain death caused by heavy cerebral trauma. All procedures for obtaining the pancreas from the donor were approved by the Human Organ Donation Ethics Committee of the affiliated hospital of Qingdao University. Informed consent was obtained from the immediate family member of the donor. This information has been added to the revised manuscript (line 112-117). We have also provided the procedure to isolate human islets in the revised manuscript (line 130-138).

2) Under different conditions not shown in this manuscript, the authors apparently do see significant effects on apoptosis in response to modulation of miR-21. Since the shown results only demonstrate islet cell viability and don't address beta cell apoptosis, it would still be possible that the observed effects are due to an increase in insulin producing beta cells rather than increased insulin secretion per se. To address this issue, it would be helpful to at least also show islet insulin content and fractional insulin release.

Response:

We thank the reviewer for this excellent suggestion. In order to exclude the possibility that the observed effects in the manuscript are due to an increase in insulin producing beta cells rather than increased insulin secretion per se, we examined the glucose-induced fractional insulin release for the following mouse pancreatic islets: 1) miR-21-efficient and deficient islets; 2) islets with or without miR-21 over-expression. Our results showed that, while miR-21 deficiency and miR-21 over-expression didn't affect total insulin content (media plus islet extract), islets from miR-21 β KO mice exhibited reduced fractional insulin release (Figure 2F) and adenovirally mediated elevation of miR-21 in mouse pancreatic islets significantly increased fractional insulin release (Figure S10). We have also stated this in the revised manuscript (line 372-377 and line 531-535).

3) The issue of therapeutic translatability has not been sufficiently toned down and there are still misleading statements raising questions about the overall understanding of the problems at hand. For example line 603-606 “Because of its high safety and effectiveness, adeno-associated viral (AAV) mediated delivery system has been widely used to treat various diseases, including diabetes. For example, AAV-mediated over-expression of glucokinase and insulin can significantly reduce blood glucose level in diabetic dog.” is highly problematic as AAV mediated systems are not being used to treat human diabetes and are not appropriate for targeting pancreatic islets as they predominantly target the liver, which was probably the case for this unreferenced dog example.

Response:

We understand the concern of the reviewer. In the revised manuscript, we have done our best to address this issue by either rephrasing or deleting statements in the Discussion regarding the therapeutic translatability of our current study.

Reviewer #5 (Remarks to the Author):

In this manuscript, Liu et al researched the role of miR-21 in mouse pancreatic beta cells using mice with beta-cell specific deletion of miR-21 (miR-21 β KO). They found that these mice developed glucose intolerance due to impaired glucose-stimulated insulin secretion (GSIS) via downregulation of Glut2 expression.

Major comments:

1. Author response it maybe be related to different cell types used (mouse cell line vs primary mouse islets) or difference in the manipulation of miR-21 expression. Thus, we think author at least research the function of miR-21a in the primary mouse islets.

Response:

We have indeed investigated the function of miR-21 in the primary mouse islets using the following two strategies:

1) We isolated pancreatic islets from wild type (WT) mice, conventional miR-21-deficient mice (miR-21KO) and mice with pancreatic β cell specific deletion of miR-21 (miR-21 β KO). Islets were then cultured *in vitro* in the presence of glucose and insulin release was examined. Our results showed that miR-21-deficient islets showed a significant reduction in glucose-induced insulin release (Figure 2A & 2D).

2) We over-expressed miR-21 in the primary mouse islets and found that adenovirally-mediated elevation of miR-21 in islets significantly increased glucose-induced insulin release (Figure 7B & 7D).

2. Author response that they have overexpress miR-21 specifically in primary mouse islet β -cell *in vitro* using an over-expression system under the control of mouse insulin-1 promoter. How to inject the over-expression system into the mouse. In the tail vein, even though the carrier contains insulin-1 promoter, it cannot be specifically expressed in β cells? Author should state clearly in the manuscript.

Response:

We thank the reviewer for pointing out this issue. We discussed the disadvantage of using over-expression system under the control of insulin-1 promoter *in vivo* in the

revised manuscript (line 634-637).

3. As mentioned in the manuscript, RIP-cre mice were used for generation of β cell specific KO. It should be noted that the transgene in RIP-cre has been found to be expressed in the hypothalamus. Whether MiR-21 was different expression in the hypothalamus?

Response:

We thank the reviewer for pointing out that RIP-cre can be expressed in the hypothalamus. We examined miR-21 expression in the hypothalamus from both wide type and miR-21 β KO mice. Our results showed that miR-21 expression was also significantly down-regulated in the hypothalamus from miR-21 β KO mice, though not as pronounced as that in the islet β -cell. This result has been added to supplemental Figures (Figure S1F). This result is consistent with previous finding that "RIP-Cre" transgene has been found to be expressed at a low level in the hypothalamus. We have also stated this in the revised manuscript (line 346-349). On the other hand, since we have shown that miR-21-deficient islets exhibit a significant reduction in glucose-induced insulin release *in vitro*, we believe that miR-21 in pancreatic islet β -cell at least directly regulates islet function.

4. In the research, whether author tested off targets effects of miR-21 KO mice, please provide.

Response:

We appreciate the suggestion from the reviewer to examine the possible off-target effects of miR-21 β KO mice. To address this issue, we performed rescue experiment by over-expressing miR-21 in miR-21-deficient islets and then examined glucose-induced insulin secretion. Our results showed that over-expression of miR-21 rescued the impaired glucose-stimulated insulin secretion by islets from miR-21 β KO mice. This result has been added to supplemental Figures (Figure S5). We have also stated this in the revised manuscript (line 377-381).

5. In the "Result 1", it is shown "We then speculate that islet miR-21 expression will be up-regulated only under the condition of combined high glucose and high fat. We confirmed this hypothesis by showing that mouse islets treated with combined high glucose and high fat but not high glucose or high fat alone significantly up-regulated miR-21 expression". It is not enough to draw the conclusion unless the author provides the expression of miR-21 in glucose and/or toxicity condition *in vivo*.

Response:

We examined the expression of miR-21 in glucose and/or toxicity condition *in vivo* by showing that miR-21 expression was up-regulated in the islets from db/db mice (display both severe obesity and elevated blood glucose level) but not ob/ob mice (display severe obesity but normal blood glucose level) (Figure 1B & 1C). In the revised Figure 1, we rearranged the order in which the data are presented by showing the results from *in vitro* study first followed by results from *in vivo* study. Per the suggestion of the reviewer, we have also rephrased the sentence (line 327-328) in the revised manuscript to "we speculate that islet miR-21 expression will be up-regulated under the condition of combined high glucose and high fat but not high fat alone".

6. A part of the Western blots (Fig 6B, Fig 6D and Fig 7M and many more) are show incomplete. The author should provide unprocessed western blot images with markers as source data.

Response:

We have provided unprocessed western blot images with markers in the updated source data file.

REVIEWERS' COMMENTS

Reviewer #2 (Remarks to the Author):

No further comments.

Reviewer #5 (Remarks to the Author):

No further comment

Response to Reviewers

We are pleased to learn that both reviewers have no concerns left. We thank the reviewers for their tremendous effort in evaluating our work throughout the peer review process.

REVIEWERS' COMMENTS

Reviewer #2 (Remarks to the Author):

No further comments.

Reviewer #5 (Remarks to the Author):

No further comment